# Intercombination Transitions in the $n = 4$ Shell of Zn-, Ga-, and Ge-Like Ions of Elements Kr through Xe

**Elmar Träbert [1,2,\*], Juan A. Santana [3], Pascal Quinet [4]** 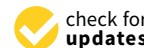 **and Patrick Palmeri [4]**

1   AIRUB, Fakultät für Physik und Astronomie, Ruhr-Universität Bochum, D-44780 Bochum, Germany
2   Physics Division, Lawrence Livermore National Laboratory, Livermore, CA 94550, USA
3   Department of Chemistry, University of Puerto Rico at Cayey, P.O. Box 372230, Cayey, PR 00737-2230, USA; juan.santana@upr.edu
4   Astrophysique et Spectroscopie, Université de Mons UMONS, B-7000 Mons, Belgium; pascal.quinet@umons.ac.be (P.Q.); patrick.palmeri@umh.ac.be (P.P.)
\*   Correspondence: traebert@astro.rub.de

**Abstract:** Earlier beam-foil measurements have targeted 4s-4p intercombination transitions in the Zn-, Ga- and Ge-like ions of Nb ($Z = 41$), Mo ($Z = 42$), Rh ($Z = 44$), Ag ($Z = 47$) and I ($Z = 53$). At the time, the spectra were calibrated with literature data on prominent lines in the Cu- and Zn-like ions. Corresponding literature data on the intercombination transitions in Ga- and Ge-like ions were largely lacking, which caused some ambiguity in the line identifications. We review the (mostly computational) progress made since. We find that a consistent set of state-of-the-art computations of Ga- and Ge-like ions would be highly desirable for revisiting the beam-foil data and the former line identifications for the elements from Kr ($Z = 36$) to Xe ($Z = 54$). We demonstrate that the literature data for these two isoelectronic sequences are insufficient, and we contribute reference computations in the process. We discuss the option of electron beam ion trap measurements as an alternative to the earlier use of classical light sources, beam-foil interaction and laser-produced plasmas, with the example of Xe ($Z = 54$).

**Keywords:** atomic spectroscopy; isoelectronic sequences; computations

## 1. Introduction

In the earlier periods (1960s to 1980s) of the quest for controlled thermonuclear fusion, the elements Fe ($Z = 26$) and Ni ($Z = 28$) as major components of the reactor vessel steel were expected to contribute contaminants to the fusion plasma. Moderately highly-charged ions of these heavy elements were seen as serious sources of radiation that would transport energy away from the core of a fusion plasma and thus hamper the attempt at reaching a sufficiently high temperature to initiate fusion among the heavy isotopes of hydrogen. Spectroscopic data were needed in order to monitor (and eventually suppress) these contaminants. Progress reports by researchers from the U.S. National Institute of Standards and Technology (NIST) delivered at many spectroscopic conferences reflect that spectroscopic tables of the time filled up rapidly up to Ni, but rarely held data of high charge states of heavier elements. Fusion test plasma reactor designs have grown steadily over the years in order to improve the ratio of the hot plasma volume to the wall area (considered to be a source of contamination). Because of the increasing heat load on the wall, then also Mo ($Z = 42$) was considered for shielding purposes and hence also as a likely contaminant, for which spectroscopic data were needed; and specific studies appeared in the literature. In the next design change (1990s), the walls of several large plasma devices were coated with a low-Z material (C ($Z = 6$) or B ($Z = 5$)) to limit the 'effective Z' of the plasma. At about the same time, medium-Z materials were injected into the plasma

to provide a 'radiation blanket' of ions that would be moderately ionized by the intense radiation from the plasma core, but would redistribute this energy geometrically over the wall and thus reduce the danger of hot spots and thus of wall damage. Depending on the plasma machine operating conditions, such radiation blankets could be provided by, for example, Ar, Fe or Kr, and the interest in spectroscopic information would range from low charge state ions (in the relatively cool plasma near the wall) to high charge states (contaminating the core plasma). Practically at each turn of the development path, it was found that atomic physics information on atomic structure, spectra and collision processes was sorely needed, but largely lacking.

Nowadays, tungsten (W, $Z = 74$) is being planned to serve in fusion reactor vessels because of its high melting point and its resilience to irradiation, and consequently, it will be the expected dominant high-$Z$ contaminant in the upcoming ITER (International Thermonuclear Experimental Reactor) experiment. At the same time, tungsten ions of specific charge states may serve as probes inside the fusion plasma, from offering the high-charge state balance and Doppler motion as a temperature sensor in the plasma core to low-charge state spectra signaling the conditions in the vicinity of the walls of the containment vessel. This role highlights a need for spectroscopic data on this element, and a rapidly growing body of data is being accumulated in various laboratories and databases. Thus, historically, the interest in spectroscopic data for the benefit of fusion reactor physics has shifted between several groups of elements, while leaving many others largely unexplored. The lighting industry (including the production of television screens) has had great interest in rare earth elements, but only in fairly low charge state ions. The purposeful shrinking of semiconductor structures for the manufacture of computer chips has fired a move towards lithography using bright sources of extreme ultraviolet (EUV) light; in this quest, moderately high charge state ions of mid-range $Z$ elements have moved into the focus of applied physics funding. The associated atomic structure work has targeted largely unresolved transition arrays, which is demanding in other ways than the accurate treatment of selected ion species of fundamental interest. We conclude that the use of fusion-related data (and other enterprises at other times) as a motivational guide has its benefits for funding purposes, but from a scientific viewpoint, one also needs to establish the systematics beyond some specific application-driven quests for special data.

In most conventional light sources (arcs, sparks, laser-produced plasmas, tokamaks, and so on), collisions with electrons (of a more or less Maxwellian "thermal" energy distribution) are essential for establishing the charge state distribution of the ions. Under such conditions, highly charged ions are more difficult to produce than those in low charge states. Consequently, the main body of experimental data has begun with low-charge ions and has evolved continually towards higher charge states, with few data available on very high charge state ions of high-$Z$ elements. For technical reasons, foremost the availability of heavy-ion accelerators, beam-foil spectroscopy has started out (half a century ago; for references, see [1]) with beams of low charge state ions. Eventually, this technique evolved to produce any charge state of any element, but there are only a few machines in the world that can provide ion beams at energies high enough to yield few-electron ions of the heaviest elements, and access time to such ion beams is scarce. With the advent of electron beam ion traps [2,3], the high-$Z$ range has become much more easily accessible to spectroscopic study [4]. Here, a quasi-monoenergetic electron beam interacts with a dilute cloud of stored ions. Highly-charged ions thus produced are even better trapped than those in low-charge states, and in this way, charge state distributions can be achieved that strongly favor highly-charged ions. This feature massively helps to explore the realm of sizeable relativistic and QED contributions to atomic structure.

However, this capability does not imply that all is well and done with experiment and with matching atomic structure computations. For one, the mid-$Z$ range of ions has partly been bypassed, and the atomic databases are grossly incomplete beyond, say, $Z = 36$ (Kr). Secondly, many experimental studies cover a few elements at a time, due to technical constraints of the spectroscopic equipment, the preparation and performance of light sources and financial resources. In computation, resources have expanded massively, and one wonders why so many recent

computations address only short sections of isoelectronic sequences (maximizing the number of publications instead of their usefulness?). The results of one computation rarely match the results of another for the same atomic system, which illustrates the limited accuracy of most atomic structure computations. If there is agreement, it may happen because certain effects are similarly (but not necessarily correctly) treated in both computations. Most such computations are presented without an estimate of intrinsic reliability (accuracy). The frequent comparison to other untested computations is meaningless. With computations for individual ions or for only short segments of isoelectronic sequences, there rarely is overlap among published computations. Of course, the most desirable comparison is that of the computational results to those of measurements, which are often scarce.

Experiments will not likely ever be able to provide all wanted data, but there are many cases in which measurements are significantly more accurate than computations. Practical measurements often yield complex spectra, the interpretation of which may strongly be helped by atomic structure theory and calculations. Evidently, an interplay of measurement and calculation is beneficial. Moreover, the composition of important contributions to atomic structure varies with the nuclear charge and the ion charge. At low ion charges, the interactions among the electrons dominate, whereas with increasing nuclear charge $Z$, the contributions of relativity and quantum electrodynamics (QED), as well as nuclear structure effects increase, and together, they eventually dominate. Therefore, a complete and consistent description of atomic structure by computer algorithms cannot be tested sufficiently well in just the low-, middle- or high-Z range alone (much less in small fractions thereof), but requires tests in all ranges.

Testing is, of course, most meaningfully done on systems that can be well measured and reliably calculated. These would be, for certain levels and transitions, relatively simple systems, that is atomic systems with one, two, three or four electrons in total. Complexity raises the stakes for atomic structure computations, but not necessarily so for experiments. We have selected a group of available wavelength measurements (in the extreme-ultraviolet wavelength (EUV) range) on ions with very few electrons outside a closed-shell electron core. We discuss a range of elements in the middle of the range of natural elements, say from Kr ($Z = 36$) to Xe ($Z = 54$) (an arbitrary choice in the present context), in order to test the accuracy of certain atomic structure calculations versus a systematic progression of complexity. For this task, we concentrate on the isoelectronic sequences of Zn, Ga and Ge, that is on ions with a Ni-like closed $3d^{10}$ shell core and 2, 3 or 4 $n = 4$ electrons in the valence shell (there are several comparable studies of lighter ions that we allude to below, with references; for further references see, for example, [1]). We mention the Cu-like ions (with a single electron outside the Ni-like electron core) only in passing, because their resonance lines have been measured, as well as treated by theory exemplarily well already [5,6], and thus, they can serve as references.

The spectra of the ions in these three isoelectronic sequences feature resonance and intercombination transitions, that is $\Delta n = 0$ electric dipole (E1) transitions to the ground state (or ground configuration) from the lowest terms of either the same spin as the ground state or a different spin value (plus opposite parity). Atomic structure computations of the ground state bank on a variational principle, according to which any imperfect wave function results in a calculated ground state energy that is higher than the true one. Thus, the lower the computational result, the better it is assumed to represent the ground state. The variational principle has been extended (less strictly) to the lowest levels of a given symmetry, which might encompass the lowest level that by a simple spin change (or multiplet mixing) can decay to the ground state; the primary example of an intercombination transition. Thus, the easy excitation of the (first) resonance transition with the excited state belonging to the same Layzer complex as the ground state often produces a particularly bright spectral line most recognizable in a given spectrum. In contrast, the lower transition rate usually lets the corresponding intercombination lines appear less brightly, but in principle, their wavelengths ought to be calculable with even higher accuracy, because the upper level involved is lower than that of the resonance line.

This problem has been studied in lighter ions before, where beam-foil spectroscopic data on Mg-, Al- and Si-like ions (all with $n = 3$ valence electrons outside a Ne-like electron core) have led to the identification of the intercombination lines of highly-charged Fe ions in the EUV spectrum of the solar corona [7,8]. It was also recognized that the predictive power of representative computations suffers from the increased complexity of the atomic system. Similar systematics were found with Zn-, Ga and Ge-like very heavy ions mostly produced in electron beam ion traps [9–11]. Highly accurate measurements and calculations on the Cu- and Zn-like ions (one or two $n = 4$ valence electrons) exist [5,6,12,13] and can be used as reference data. Based on our experience, it appears advisable that any new computations on Ga- and Ge-like ions demonstrate their capability by application to the Zn isoelectronic sequence first.

The similarity of Zn-, Ga- and Ge-like ions with lighter ions of a closed-shell core and just a few valence electrons is deceptive, of course. In Zn-, Ga and Ge-like ions, the valence electrons have angular momentum quantum numbers largely similar to those in the core, and the concept of an inert electron core surely is being stretched. In practice, the present-day atomic structure computer packages therefore treat thousands of inner excitations of core electrons by employing many thousands, if not millions, of basis wave functions.

There is a fair number of spectroscopic studies that have addressed Cu-, Zn-, Ga- and Ge-like spectra in the atomic number range of present interest. These studies (examples are cited and discussed below) have used a variety of experimental techniques and have produced data on many spectra and term systems. Upon closer inspection, however, one finds that the intercombination transitions have rarely been reported. This can have many reasons; the lines are usually weaker than the principal resonance line, and the particle density in the light source plays a role, as well, because longer-lived levels might be quenched collisionally. Interestingly, the beam-foil light source (with its frequent low-signal rate problems) offers particular features that are actually helpful. In observations of the ion beam at some time after excitation, the relative signal of "slow" transition can be enhanced, because the "fast" transitions die out first (while the total signal rate decreases, nevertheless). For five elements from Nb ($Z = 41$) to I ($Z = 53$), beam foil experiments have reported intercombination transitions in Zn- to Ge-like ions. While the results for Ga-like ions largely match the expected wavelength and line intensity pattern, the data for Ge-like ions are less conclusive. In this report, we discuss the situation and review the evidence from these five experiments combined. We also look at an EUV spectrum of Xe ($Z = 54$) from an electron beam ion trap and try to establish corresponding information.

Short of more measurements for more elements, it seems appropriate to intercompare the available measured data along the isoelectronic sequence. For such an enterprise, it would be good to have reliable computed data or at least isoelectronic trends that could be adjusted by reference to measurements. We survey the literature (and some unpublished work of our own) in order to identify such computations and to scrutinize experimental and computational results in the literature in terms of consistency. It is beyond our task horizon to analyze the individual atomic structure computational approaches for their details, weaknesses or strengths. The present study therefore does not present a new idea, but applies the essence of the earlier findings to a specific sample of beam-foil spectroscopic and electron beam ion trap data, in order to locate the frontier of reliability of such data and the associated spectrum interpretation.

## 2. Available Data

Spectroscopy using classical light sources has been active for more than a century. Highly-charged ions emit mostly in the X-ray and EUV spectral ranges. The development of, both, spark discharge light sources and grazing incidence spectroscopy by Bengt Edlén and his group at Uppsala in the 1930s has opened up wide avenues of research. Thus, it has become possible to study all spectra of a given element, as well as to analyze isoelectronically-similar spectra of a sequence of elements. When in the 1950s and 1960s tokamak plasmas and laser-produced plasmas began to produce highly-charged ions, Edlén's toolbox [14] (based on series expansions of atomic parameters as developed by Egil

Hylleraas since the early 1930s) was readily available. Subsequently, sounding rockets and space flights have transported EUV and X-ray spectrometers beyond Earth's atmosphere, and observations of the rich solar corona spectra have added to the interest in the spectroscopy of all elements and many charge states. Data from many enterprises began to be compiled and stored institutionally. Looking at such compilations, one may feel overwhelmed by all that accumulated knowledge, especially as some compilations claim to cover "all" spectra of a given element (in the *Z* range of present interest, there are, for example, the studies by Sugar et al., Shirai et al. and by Saloman [15–20]). However, when looking up the entries for Ga- and Ge-like ions in these compilations, the actual scarcity of data becomes evident. Fine structure intervals in the ground configuration often are reported from work using Cowan code computations (based on the long-standing and well-established pseudo-relativistic Hartree–Fock approach, a method of atomic structure modeling that can be used with adjustable radial energy parameters), aiming to bridge the gaps between observations. These semi-empirically adjustable computations are helpful for the practitioner, but they need experimental data as anchors. Atomic structure theory is highly developed, but its implementation by computation, preferably using *ab initio* approaches, leaves much to be desired (see discussion below).

Isoelectronic analyses following Edlén's example are a useful tool to spot various errors. Of course, nowadays, on-line databases such as NIST ASD [21] can add new data more frequently than olden-day print cycles of paper-based journals of reference would permit, but for the atomic systems of present interest, not much has changed. Moreover, in certain aspects, not much can change. For example, since the late 1970s, laser-produced plasmas and tokamak experiments have addressed the resonance lines in Cu- and Zn-like ions of many elements (see, among others, [22–29] for mostly Cu-like ions and [30–44] for mostly Zn-like ions), often with NIST personnel as collaborators, and the results have been adopted by the NIST table of reference compilation group. Any extension to other parts of the same atomic systems would require cross checks and possibly re-calibrations, or the tabulations would become inconsistent. An example is the $4s^2\ ^1S_0$–$4s4p\ ^3P^o_1$ intercombination transition in the Zn-like ion (discussed below) in which the NIST ASD online database has a level value for Br VI ($Z = 35$) that differs significantly (by some $700\ \mathrm{cm}^{-1}$) from the isoelectronic trend of many later measurements. However, the NIST ASD data entry in the 2018 version is part of a comprehensive term analysis of Br VI that had been entered into the (U.S.) National Bureau of Standards database in the early 1950s, probably by Charlotte Moore Sitterly, who in those years practically single-handedly set up the corner stones to that monumental reference collection. The entries for Br VI include the three fine structure levels of the $4s4p\ ^3P^o$ term and their relative positions close to the well-established interval rule. Without corresponding new measurements of many levels and a subsequent new term analysis, the (assumed) internal consistency of not only this term, but of the existing dataset would be lost by correcting just that particular $J = 1$ level. Such a single-entry correction of the database would be counter-productive, but one has to be aware of the occurrence of such problems.

Most wavelength measurements using the already mentioned beam-foil technique (a beam of fast ions passes through a thin foil, usually made of carbon, and is being excited by the interaction with this solid-state density electron target) are less accurate than tokamak or laser-produced plasma measurements, because the Doppler effect in the observation of fast ions (with a speed on the order of a few percent of the speed of light) causes a variety of problems. However, the experiments are intrinsically time-resolved (for references, see, for example, [1,45]), and this feature offers various interesting options. For one, atomic level lifetimes in the range from a few picoseconds to many nanoseconds can be measured. Secondly, wavelength measurements can be tried by viewing the ion beam near the exciter foil (where many core-excited level decays contribute to a line-rich spectrum dominated by the decays of short-lived levels) or downstream of the foil (where fast decays have largely died out and the decays of long-lived levels are relatively more prominent) (see [7,8,46]). Concerning the Cu- through Ge-like ions of present interest, beam-foil spectra have been obtained at the Bochum Dynamitron tandem accelerator laboratory on the five elements Nb, Mo, Rh, Ag and I. The experiment used a $R = 2$ m, 600 $\ell$/mm grating grazing-incidence scanning monochromator equipped with a

Channeltron detector. The channel-by-channel data acquisition implied that in a full day of operation, maybe about 2000 channels of 30-s data accumulation time each could be covered. To achieve this, the ion source had to be productive and long-lasting; the Bochum accelerator was equipped with a high-current sputter source, and the accelerator was capable of handling multi-$\mu A$ ion currents. The exciter foil suffers from the heavy-ion impact and eventually develops small holes and even ruptures; some of the deterioration in light yield is difficult to recognize in time, especially with low-count rate measurements as in these recordings of delayed spectra.

The Bochum EUV wavelength spectra of niobium (Nb, $Z = 41$) [47–49], molybdenum (Mo, $Z = 42$) [50,51], rhodium (Rh, $Z = 45$) [52,53], silver (Ag, $Z = 47$) [50,54,55] and iodine (I, $Z = 53$) [56,57] have been calibrated with $4s_{1/2}$-$4p_{1/2,3/2}$ reference lines in Cu-like ions and the $4s^2$ $^1S_0$–$4s4p$ $^{1,3}P_1^o$ resonance transitions in the Zn-like ion (there are also beam-foil lifetime measurements of the resonance lines of Cu-like iodine (experiments done at Brookhaven [58]), and several of the above studies included measurements of the $4s^2$ $^1S_0$–$4s4p$ $^3P_1^o$ intercombination transition rate in the Zn-like ion, providing another test of the quality of atomic structure computations). The wavelength scale thus established was then used to identify the nearby components of the $4s^24p$ $^2P_{1/2,3/2}^o$-$4s4p^2$ $^4P_{1/2,3/2,5/2}$ intercombination line multiplet in the Ga-like ion and the $4s^24p^2$ $^3P_{1,2}$-$4s4p^3$ $^5S_2^o$ intercombination line doublet in the Ge-like ion. These transition arrays are (separately) discussed below. The three-decades old data of the studies cited above have been used here again to produce sample spectra with similar dispersion (Figures 1–3) in order to visualize the measurement situation of the present physics case.

The aforementioned beam-foil data have been measured at tandem accelerators, which because of their design principle, cannot operate with rare gases beyond He. Hence, Xe is beyond the reach of these machines, but easily available at electron beam ion traps. In earlier experiments at the Livermore electron beam ion trap, the electron beam energy was chosen high enough to produce three- and four-electron ions of Xe, and the same high-resolution EUV spectra also showed lines of Xe ions with three and four electrons outside a Ne-like electron core, that is Al- and Si-like ions [59]. At the Berlin Electron Beam Ion Trap (BEBIT), Biedermann et al. [60] varied the electron beam energy systematically so that from one spectrum to the next, preferably a single charge state was added to (or removed from) the upper end of the charge state distribution in the trap. This technique had been exploited for various elements, especially Fe, at the Livermore EBIT laboratory before [61,62]. Given the multitude of charge states present among the trapped Xe ions and the many multiplet overlaps and line blends in the EUV spectrum, such a systematic variation can help with disentangling the spectra and thus identifying the lines. Concerning the present context, Biedermann et al. [60] reported lines of Cu- to Rb-like spectra observed below 25 nm with moderately high resolution. Electron beam ion trap laboratories worldwide boast spectrographs (mostly of the flat-field type) with multichannel detection (the first electron beam ion traps came into operation a quarter of a century after beam-foil spectroscopy began). Consequently, a typical electron beam ion trap spectrum of 30-min exposure implies that each channel had 30-min data accumulation time simultaneously, which explains in part why the signal count numbers are much higher than in those beam-foil spectroscopy examples that were recorded with equipment from before the advent of multi-channel detection (see Figure 3). With position-sensitive detectors in beam-foil spectroscopy (see [46,63]), the difference is less drastic.

Figure 3 shows another Xe spectrum from the Livermore electron beam ion trap EBIT-I. This spectrum has been recorded with an electron beam energy of 1000 eV, just above the ionization limit of the Cu-like Xe ion. Thus, Ni-like ions may be present as the highest charge state in the trap, but only as a small fraction of the charge state distribution. Cu- and Zn-like ions are expected to be most prominent and Ga- and Ge-like ions somewhat less so. However, since the electron beam ion trap was operated with a continuous influx of neutral Xe gas (under high-vacuum conditions), as well as cycled every so often to remove unwanted heavy-element contaminants, all lower charge states of Xe are expected to be present, but in small amounts. This spectrum has been dispersed with the moderate resolution LoWEUS spectrograph (Long-wavelength Extreme Ultraviolet Spectrometer) [64,65], covering a range roughly from 20 to 45 nm, which extends the

study by Biedermann et al. [60] considerably in wavelength, but (so far) without the systematic variation of the electron beam energy. The shortest-wavelength (unidentified) line at 21.118 nm is also seen in second order at 42.230 nm, and so are the mountains of blended lines at the adjacent longer wavelengths. Several lines seen by Biedermann et al. [60] appear in this spectrum, as well. The resonance line $4s_{1/2}$-$4p_{3/2}$ in Cu-like Xe is seen in second order, the $4s_{1/2}$-$4p_{1/2}$ line in first order. Correspondingly, the resonance line $4s^2\ ^1S_0$–$4s4p\ ^1P^o_1$ in the Zn-like ion of Xe is seen in second order of diffraction, while the intercombination transition $4s^2\ ^1S_0$–$4s4p\ ^3P^o_1$ appears in first diffraction order as one of the strongest lines in the spectral range. These four lines corroborate the wavelength calibration, which for this spectrograph setting had been established with reference lines in the spectra of low-Z elements.

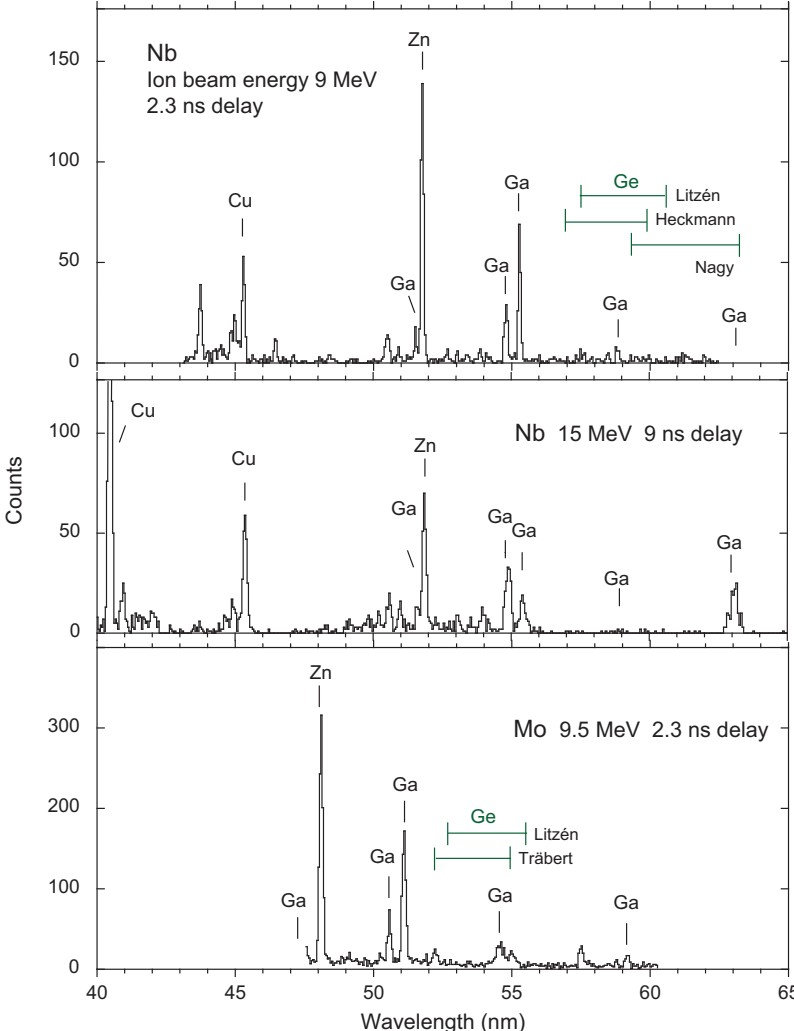

**Figure 1.** Delayed beam-foil spectra of Nb and Mo in the wavelength range of the Zn-, Ga- and Ge-like ion intercombination lines. Prominent lines are marked by the isoelectronic sequence; the five Ga-like ion intercombination lines are indicated at positions stated by Biémont et al. [66], the two intercombination lines in Ge-like ions (green markers) at positions claimed by experiment [48,51,67] or theory [68]. The Nb spectrum in the middle panel has been recorded farther away from the exciter foil (at a longer delay after excitation); therefore, wider spectrometer slits were used, which result in a larger line width. Changes in relative line intensity between the spectra relate to differences in upper level lifetime. Such spectra are also affected by the deterioration of the exciter foil under ion beam impact. The limited foil standing time and available machine time force the experimenter to economize by tailoring the measurement program, with the result that not all the lines of interest may be covered in such a spectroscopic scan.

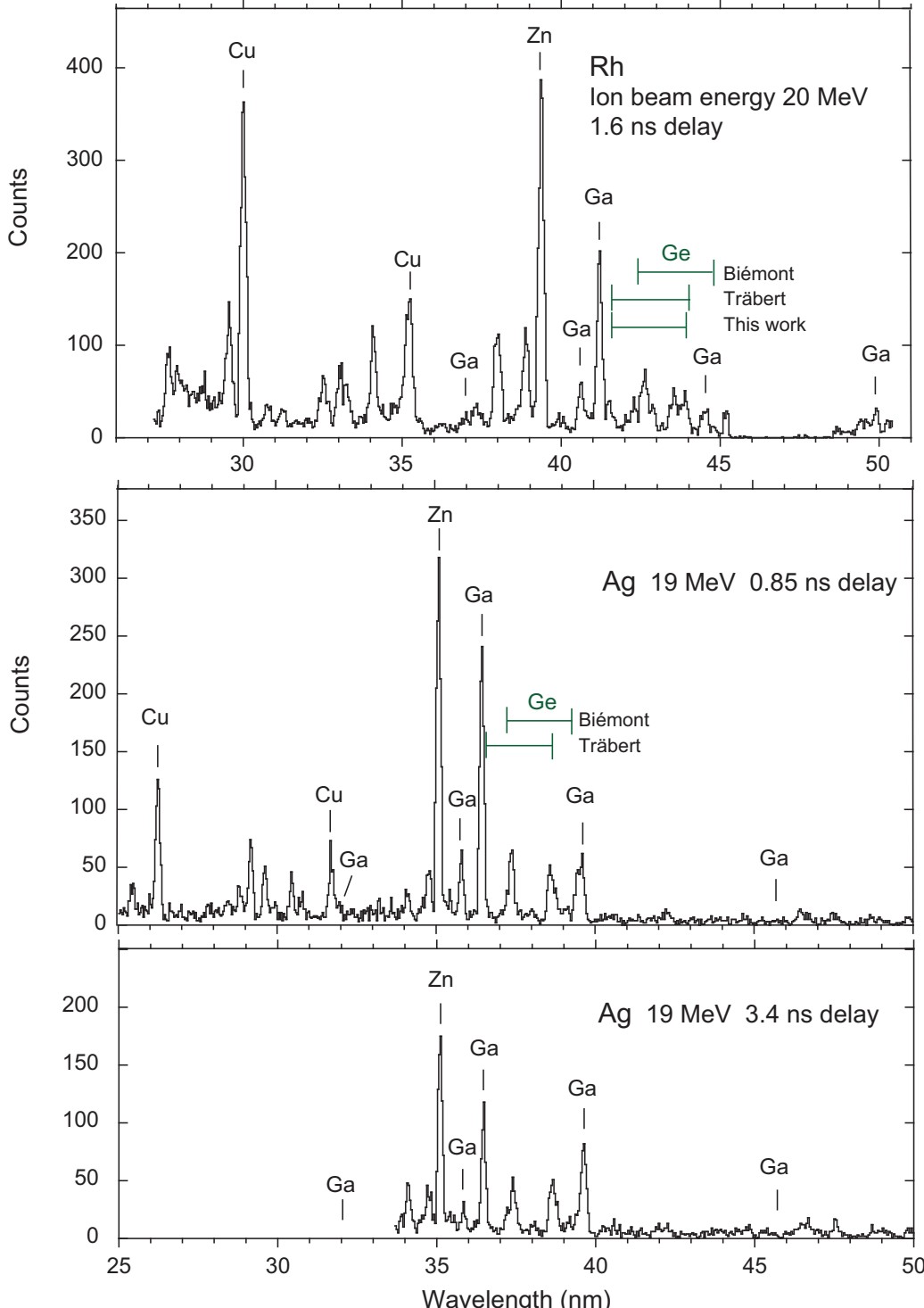

**Figure 2.** Delayed beam-foil spectra of Rh and Ag in the wavelength range of the Zn-, Ga- and Ge-like ion intercombination lines. Prominent lines are marked by the isoelectronic sequence; the five Ga-like ion intercombination lines are indicated at positions stated by Biémont and Quinet [69], the two intercombination lines in Ge-like ions (green markers) at positions claimed by experiment [52,55] or theory [70].

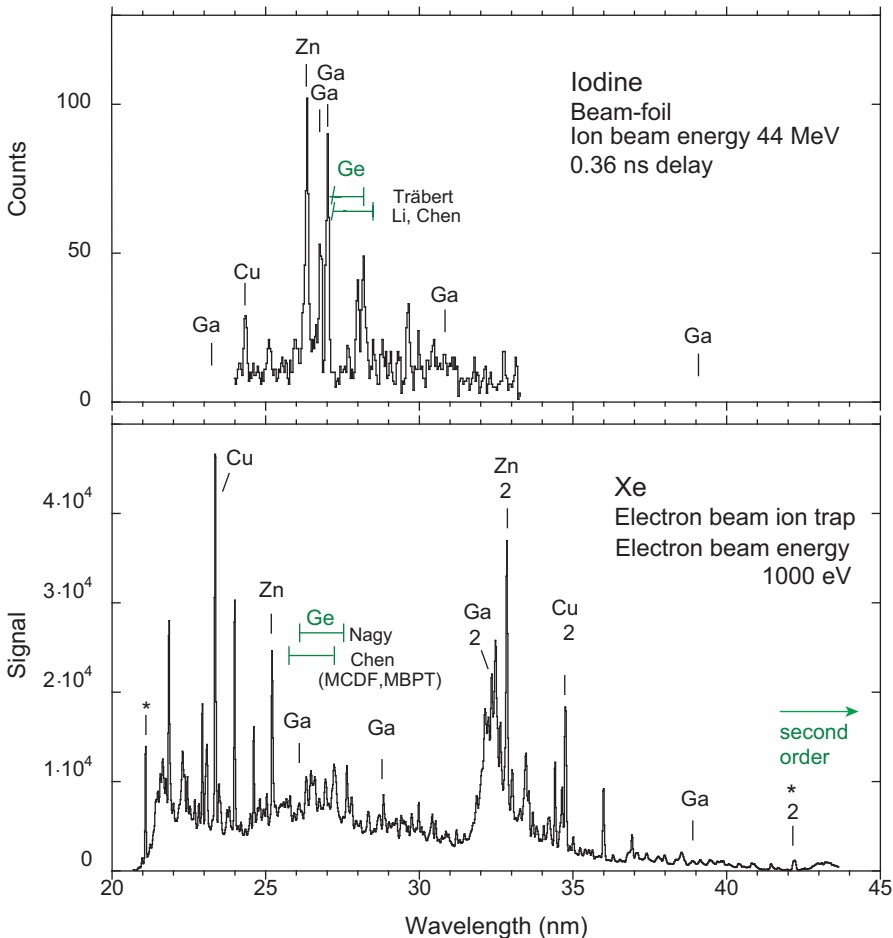

**Figure 3.** Delayed (Bochum) beam-foil spectrum of iodine [56] (upper panel) and (Livermore) electron beam ion trap spectrum of xenon (lower panel, [71]) in the wavelength range of the Zn-, Ga- and Ge-like ion intercombination lines. The line labels indicate isoelectronic sequences and appearance in second order of diffraction. At the long-wavelength end of the Xe spectrum, the short-wavelength range of the same spectrum appears to be shown in second order; see the line marked with an asterisk (*). The beam-foil spectrum has been measured with a scanning grazing-incidence monochromator equipped with a Channeltron detector; the electron beam ion trap measurement employed a flat-field grazing-incidence spectrograph equipped with a cryogenic CCD camera. The latter measurement required only about a tenth of the recording time of the former. The green double markers point to the two intercombination transitions in the Ge-like ions as suggested by experiment [56] or theory [57,68,72]. Neither pointer is unambiguous.

Our spectrum analysis fit program recognizes about 90 lines in the range depicted in Figure 3 (bottom panel). Visual inspection suggests the presence of a fair number of additional lines, mostly weak ones, and of various blended components. These might be analyzed meaningfully only when the presently structured background can be resolved into its very many constituent lines. In part, this is a technical problem asking for a higher-resolution instrument, but the identification also needs extensive theoretical support. Biedermann et al. [60] used the HULLACcode [73] to obtain computational information on the spectral structure; their results from this procedure scatter by about 1% around their experimental wavelength data. Within such an uncertainty interval, the EBIT-I data in most cases have several candidate lines, and only for the strongest lines in a few spectra, the association with prediction is straightforward. With more than two electrons in the valence shell, the uncertainty interval of the computational results increases markedly, and thus, the predictive power rapidly decreases.

However, there is another complication with EBIT data in the present context. The two spectra in Figure 3 demonstrate a significant difference of beam-foil and electron beam ion trap spectra, which in this case disfavors the latter. The EBIT comprises a low-density plasma, whereas the interaction of fast ion beams with solid-state foils occurs in a high-density environment. In the latter, excitation to many levels is efficient, and cascade effects afterwards feed the resonance lines so that they stand out by their line intensity, as do some of the feeding transitions. Among the lower-lying levels, the statistical weight $(2J + 1)$ is an important factor in the level population under high-density (high-collision frequency) conditions, and multiple excitation is quite normal. Recently, it has been shown how the different level populations of a higher lying level affect the brightness of resonance lines (in that case, of Fe spectra [74]), because the cascade repopulation then is drastically different, high after ion beam-foil interaction and very low in an electron beam ion trap.

At low density (low collision frequency), the likelihood of radiative decay is higher than that of collisional excitation. Consequently, the dominant fraction of ions are in the ground state, and the appearance of transitions in the spectrum is heavily weighted towards those upper levels that are excited easily (and directly) from the ground state. A telling example are intercombination transitions in ions with three valence shell electrons (in our case, Ga-like ions; see Al-like ions for similar effects [75]). In beam-foil spectra, the strongest line of the $4s^2 4p\ ^2P^o_J$-$4s4p^2\ ^4P'_J$ intercombination transition multiplet is the $J = 3/2 - J' = 5/2$ component, massively favored by statistical weight over the neighboring $J = 1/2 - J' = 1/2$ component. This prominent line is thus easy to recognize in the vicinity of the $4s^2\ ^1S_0$–$4s4p\ ^3P^o_1$ intercombination line of the Zn-like ion. In the electron beam ion trap, the $J = 3/2$ fine structure level of the ground term is much less populated than the $J = 1/2$ ground level of the same term, and consequently, the $J = 3/2 - J' = 5/2$ component of the intercombination line multiplet is much weaker. In a Au spectrum from the Livermore EBIT-II electron beam ion trap [75], the line intensity pattern of Al-like ions actually does not show the $J = 3/2 - J' = 5/2$ prominently, but the $J = 1/2 - J' = 1/2$ transition instead (that in beam-foil spectra is weak), because here, the lower level is the well-populated ground state.

Indeed, in the Xe spectrum from the Livermore EBIT-I, there is no such very strong line in the same brightness class seen as the $4s_{1/2}$-$4p_{1/2}$ lines of the Cu- and Zn-like ions (Figure 3). For hints at which of the numerous moderately bright lines with a wavelength near 26 nm might be the correct one in the Ga-like ion, we have two possible pointers: one would be theory, but apparently, there is no sufficiently accurate prediction in the literature; the other would be the isoelectronic trend, but there is only one measurement nearby (for iodine, $Z = 53$, [56,57]), and no sufficiently reliable isoelectronic scaling has been provided by theory yet either.

## 3. Discussion of Wavelength Data on Cu- through Ge-Like Ions

### 3.1. Cu-Like Ions

There are two major calculations of highly-charged Cu-like ions in the literature, one by Kim et al. [5] and another one by Blundell [6], both treating atomic systems with one valence electron outside a closed-shell core, with results for Li- ($1s^2$ (He) core), Na- ($2p^6$ (Ne) core) and Cu-like ($3d^{10}$ (Ni) core) ions. Accurate measurements at very high values of $Z$ (mostly done on the $ns_{1/2}$-$np_{3/2}$ transition of ions stored in electron beam ion traps; see [9,11]) give a slight preference to the former computations (which involve some semi-empirical adjustment in the QED contribution) over the latter (which is *ab initio* throughout). However, the actual situation is less clear-cut than it might appear. For example, the $4s_{1/2}$-$4p_{1/2}$ transition is weaker than the $4s_{1/2}$-$4p_{3/2}$ transition, and in the different wavelength ranges, the measurement accuracy has achieved different heights. Here, we discuss only the $4s_{1/2}$-$4p_{1/2}$ transition, because the $4p_{1/2}$ wave function is also involved in the intercombination transition of the ions with additional electrons in the valence shell.

The computations by Kim et al. [5] cover all elements, which makes it easy to compare experimental results. In contrast, the Blundell computation treats only a few elements, which are not necessarily

the ones amenable to the best measurements. When intercomparing the two sets of computed results (see Figure 4), there are evident changes along the isoelectronic sequence that one would like to know more about, for example whether the deviations follow smooth trends or occur for a few particular ion species, but the intermittent Blundell coverage does not permit such an analysis. However, this is a complaint at a high comfort level, since for most ions (and the $n_{1/2}$-$n_{1/2}$ transition), the two sets of results differ by only about 200 ppm from each other, while originally, few experiments would reach such a high level of accuracy. By the way, neither theoretical treatise actually specifies an estimate of accuracy.

　　If one compares the computed results with the experimental data since collected (Figure 4), one finds that there is an offset of about 100 cm$^{-1}$ between the results obtained by Kim et al. [5] and the measurements (in a wide range of $Z$), whereas some of the (few) Blundell results [6] differ by less than that from measurement, but without a clear trend. Thus, for some elements, the absolute mismatch is smaller here, but because of the sparsity of the computed results, one would like to have information for more elements. In contrast, the offset of the computed results obtained by Kim et al. from experiment is so constant that the authors add this offset to their computed results and thus obtain "predicted" values, which are very close to the experiment, indeed often with deviations of only 10 to 50 ppm. The offset has not yet been explained from first principles, but for practical purposes, it permits excellent predictions (guided by the body of experimental data).

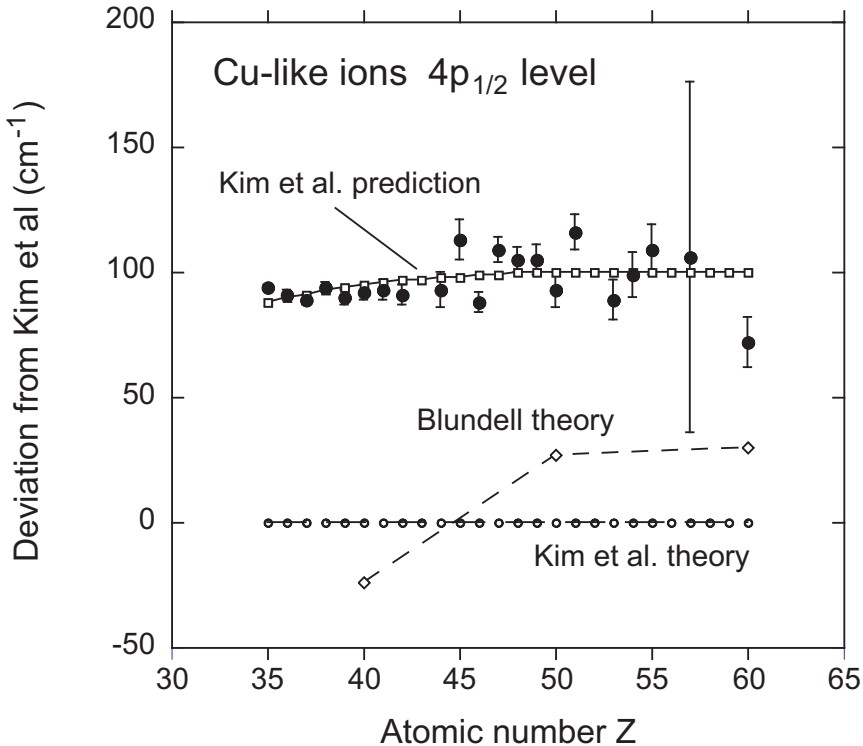

**Figure 4.** Deviations among calculations by Kim et al. [5] and by Blundell [6] and measurement for the 4p$_{1/2}$ level in Cu-like ions.

We note that the above computations are, of course, not the only ones. For example, Seely et al. and Kania et al. [28,29,35] have done semi-empirically-guided computations to match their laser-produced plasma measurements of Cu- and Zn-like ions, and Palmeri et al. [76] have performed computations for Cu-like ions of elements $Z = 70$ and above. However, at very high $Z$, the results of one set of computations deviate from our reference set by 1000 cm$^{-1}$ and more to the one side, and the other set deviates to the other side. These high atomic numbers lie beyond our present range of interest and are therefore not included in our figure.

### 3.2. Zn-Like Ions

The $4s_{1/2}$-$4p_{3/2}$ transition in Cu-like ions corresponds to the $4s^2$ $^1S_0$–$4s4p$ $^1P_1^o$ resonance transition in Zn-like ions and is of less concern here than the $4s_{1/2}$-$4p_{1/2}$ transition in Cu-like ions, which corresponds to the $4s^2$ $^1S_0$–$4s4p$ $^3P_1^o$ intercombination transition in Zn-like ions (and similarly, to the intercombination transitions in Ga- and Ge-like ions). The intercombination line in the Zn-like ion has been observed and measured in many elements for decades. Instead of referencing the original measurements, we rely on the NIST ASD database [21], a systematic data analysis by Churilov et al. [41] and measurements (supported by relativistic computations) of laser-produced plasma spectra by Brown et al. [44]. However, on the computational ("theory") side, there are so many attempts at delivering a good description of the major transitions in Zn-like ions that they may serve as illustrative samples of different degrees of success (see Figure 5), even without delving into details of the theoretical approaches used.

In the 1980s, several light sources provided access to the study of highly-charged Zn-like ions, such as laser-produced plasmas, tokamak discharges and foil-excited ion beams. Responding to a need, Curtis [77,78] elected to systematize the wavelength and transition rate data on Zn-like ions, largely by a semi-empirical description, which he used to extend the isoelectronic trends from the measured data at low $Z$. These predictions were useful at the time; two decades later, his level energy extrapolation to $Z = 54$ was found to fall short of measurement (compiled at NIST [20]) by about one percent. Moreover, the isoelectronic trend curved away from the growing body of experimental evidence (see Figure 5a,b). Around the same time, a fully-relativistic computation, using the multi-configuration relativistic random phase approximation (MCRRPA) dared to cover the full isoelectronic sequence [79]; this meritful attempt, however, had limited success (see Figure 5a). Within the subsequent two decades, several other computations yielded isoelectronic trends that in the end did not agree with the experiment either (see Figure 5b). All of these computations covered a section each of the isoelectronic sequence, which may cause problems for the user in need of reliable estimates of atomic data and in quality assessment. For example, two (relativistic) multi-configuration Dirac–Fock (MCDF) calculations by Biémont claim a coverage from Rb VIII ($Z = 37$) to Xe XXV ($Z = 54$) or W XLV ($Z = 74$) in the abstract [80,81], but in both cases, the listed results actually begin at $Z = 47$ (Ag). A later MCDF computation by Quinet et al. [82] begins at $Z = 70$ (and extends to uranium ($Z = 92$); it thus complements the earlier work. It is very laudable that the calculations (performed by members of the same team) do overlap, indicating how different (but related) atomic structure packages yield somewhat different results. In the range of overlap, the $4s4p$ $^3P_1^o$ level energies of the two sets of calculations differ by some 1400 ppm from each other. The later computations evidently yield a more suitable (but not yet satisfactory) high-$Z$ data trend than the earlier ones. Of course, a meaningful estimate of the accuracy of the computational approaches and results would be highly appreciated by any users of the results. Liu et al. [83] have produced results mostly for relatively low-$Z$ elements. There is another set of results that has been obtained by Vilkas and Ishikawa [84] that seems to oscillate around the reference dataset (in our figure), the relativistic configuration interaction (RCI) computations by Chen and Cheng [12] (see this paper also for a detailed discussion of the theoretical problems and achievements plus more references). Vilkas and Ishikawa employed a multi-reference Moller–Plesset (MRMP) approach; after further development work on this code and its implementation, Santana [13] has obtained results without such wiggles and actually fairly close to the RCI results (but on the "far" side of experiment) for the whole length of the isoelectronic sequence (see Figure 5c). Two more computations have produced results in the same ballpark; these are the relativistic many-body perturbation theory (RMBPT) exercises by Blundell and the Safronovas [85–87]. Their results come closest to the experimental data, and their trends even cross near $Z = 50$. At higher $Z$, beyond the range depicted in our figure, improved-QED computations by Blundell agree better than the (basically similar) Safronova computations with experimental data (on the $4s4p$ $J = 3/2$ level) from the Livermore electron beam ion trap (see discussions in [12,88]). Nevertheless, the computations by Chen and Cheng turn out to be superior also in the high-$Z$ range.

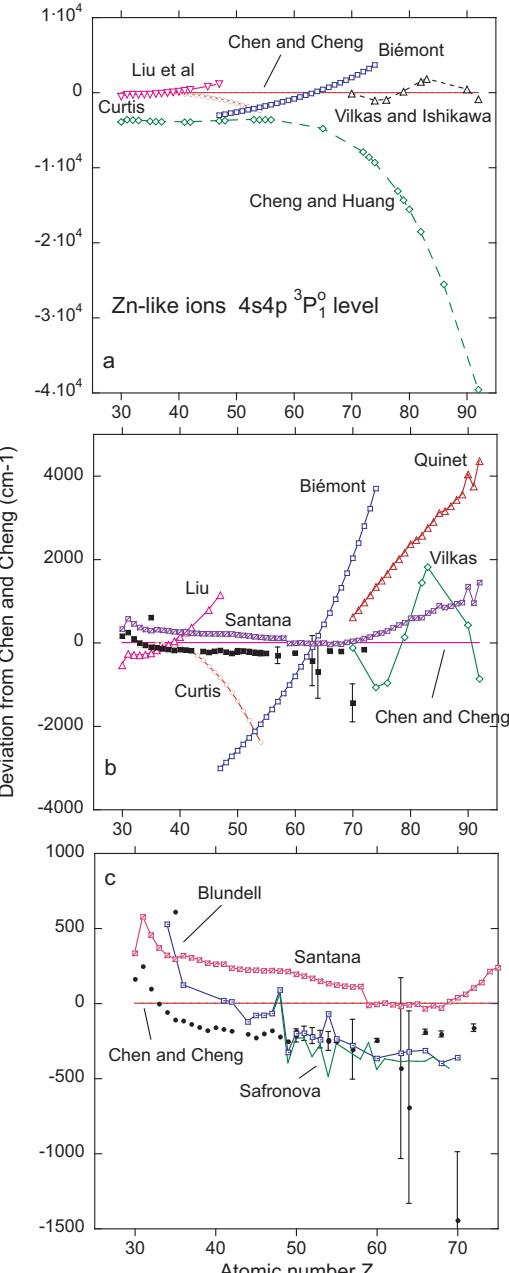

**Figure 5.** Deviations among calculations and measurement for the $4s4p_{1/2}\ ^3P_1^o$ level in Zn-like ions. All entries are shown relative to the results of the relativistic configuration interaction (RCI) computations by Chen and Cheng [12] (magenta line at zero). The experimental data (full black symbols) have been taken from the compilations by NIST [21], by Churilov et al. [41] (both without error bars) and from the experiment by Brown et al. [44] (with error bars). Note that for $Z = 35$, the NIST online database has an entry that is clearly off the trend of all other experiments. (**a**) (Top panel) The emphasis is on the oldest of the computations. There are large deviations especially at high $Z$. (**b**) (Middle panel) The poorest computations of (**a**) have been replaced with some newer ones and the vertical scale expanded. Note how the isoelectronic trends of some computations deviate from the experimental data. (**c**) (Bottom panel) Only the computations with the smallest deviation from experiment are shown. Key to computations: Curtis [77], Biémont et al. [80,81], Cheng and Huang [89], Vilkas and Ishikawa [84], Liu et al. [83], Quinet et al. [82], Blundell [85], Safronova et al. [87], Santana [13].

There also are computations that address individual elements (for some Zn-like ions, see, for example, [90,91]), but these do not help with learning about isoelectronic trends. Fortunately, for Zn-like ions, there are by now results of computations available that span the whole isoelectronic sequence with its many elements [12,13]. In the $Z$ range of present interest (Kr to Xe), the results of these two computations differ by about 200 cm$^{-1}$ from each other and from experiment, which is excellent. However, because of the underlying atomic structure energy scaling, this difference translates to almost three thousand ppm near Kr (not so impressive) and to about 300 ppm near Xe. The experiment differs from our reference computation [12] by about one thousand to six hundred ppm (with a sign opposite of that for the other recent computation [13]). Evidently the best of the computations for Zn-like ions (two valence shell electrons) has reached a high accuracy, but this accuracy is notably poorer than that achieved for Cu-like ions (one valence shell electron).

### 3.3. Ga-Like Ions

The low-lying levels in Ga-like ions feature fine structure splittings that scale roughly with $Z^4$ and a 4s$^2$4p $^2$P$^o$-4s4p$^2$ $^4$P term separation that scales roughly linearly with $Z$. The ground state fine structure interval grows by about a factor of 20 from Br ($Z = 35$) to Xe ($Z = 54$), and the 4s4p$^2$ $^4$P$^o_{5/2}$ level value increases by a factor of more than five over the same range. It is difficult to display such changes in graphics and still see small deviations from overall trends. In the following, we present and discuss two ways to tackle and visualize this problem, one based on measurements and semi-empirically-adjusted computations trying to extrapolate to higher values of $Z$ and the other referring to *ab initio* computations.

For the ground term fine structure, the experimental data compilation for $Z = 35$ to 49 by Curtis and Reader [92], Reader et al. [93], Litzén and Reader [94], Litzén and Zeng [95] and the semi-empirically adjusted computation by Biémont and Quinet [69] largely agree with each other; the combined dataset can be fitted by a polynomial and thus also extrapolated. For the quartet term, the computation by Biémont and Quinet [69] reaches up to $Z = 49$; their results can be augmented by other computational results for $Z = 35$ [96] (from MCDF calculations for Ge II through Nb XI)), $Z = 47$ [91] and $Z = 53$ [57]. The measurements, data compilations and extrapolations by atomic structure calculations (using the Cowan code with adjustable parameters) by Litzén, Reader and Zeng [94,95] reach from Rb ($Z = 37$) to In ($Z = 49$); they are reproduced by (adjusted) computations [66,69] to within 0.3% to 0.7% at the high-$Z$ end of this range. In Figures 1 and 2, the positions of the five 4s$^2$4p $^2$P$^o_J$-4s4p$^2$ $^4$P$_{J'}$ multiplet components are marked according to the results of classical light source work and spectral analysis by Litzén, Reader and Zeng. Apparently, those researchers did not observe all or even most of these intercombination transitions, but they combined other accurate observations with scaled Cowan code computations to provide a complete set of level energies. The relative line intensities of the five components in the intercombination line multiplet of the Ga-like ion can be estimated from simple assumptions about initial level populations, computed level lifetimes and branch fractions, and several components can in most cases be identified with little ambiguity in the beam-foil spectra of Figures 1 and 2.

The beam-foil work in most cases is less accurate than work with stationary light sources in terms of wavelengths, but helpful in terms of phenomenology, that is the confirmation of longevity for intercombination transitions with their relatively long-lived upper levels. Indeed, many of the weaker members of the intercombination transition multiplet appear to be partly blended by other unidentified lines in the beam-foil spectra (which are isotopically pure, and thus, the blends are all from ions of the same element). By combining spectra recorded at different delays after excitation, the identity of a line of interest can often be ascertained. The same blends are evident also in decay curve measurements. There, the decays of long-lived levels (say, in the nanosecond range) are readily separated from most of the short-lived blends. Unfortunately, the longer the delay time, the lower is the signal rate, and if a spectral feature is weak to begin with (at the time the beam ions emerge from the exciter foil), poor signal statistics limit the value of any evidence. In short, the beam-foil spectra for

Nb, Mo, Rh and Ag corroborate nicely the spectral analysis of the intercombination transition array predicted by classical spectroscopy on different sets of transitions. The question arises how far an extrapolation of this consolidated knowledge might carry. For example, a fit to the measured ground term fine structure interval data (compiled by Curtis and Reader et al. [92,97], reaching up to $Z = 49$) yields a prediction for $Z = 56$ (Ba) that falls short of the (much more recently) measured value [98] by about $850\,\mathrm{cm}^{-1}$ or 0.5%. Any corresponding analysis of the components of the $4s^2 4p\ ^2P^o_J$-$4s4p^2\ ^4P_{J'}$ transition array would carry larger uncertainties, because the upper levels from which to extrapolate are less well established.

For a didactical example that is largely based on experimental data, we elect to display the fine structure splittings of the $4s4p^2\ ^4P$ levels relative to the splitting of the $4s^2 4p\ ^2P^o$ ground term in Ga-like ions (see Figure 6). The $^4P\ J = 1/2 - 5/2$ level interval of the lowest quartet term is almost as large (close to 90% and practically constant at that) as the $^2P^o\ J = 1/2 - 3/2$ fine structure splitting of the ground term. The $^4P\ J = 1/2 - 3/2$ level interval increases from roughly 40% of the ground term fine structure near $Z = 35$ to more than 60% near $Z = 49$. This presently is the practical end of the range for accurate observations of all three quartet levels, and the question arises how to reach beyond and how reliable an extrapolation of the isoelectronic trend might be. The next available measurement (on iodine) is four units of atomic charge away from the upper end of the range and has not reached an accuracy as high as that of the aforementioned studies by Litzén, Reader and Zeng.

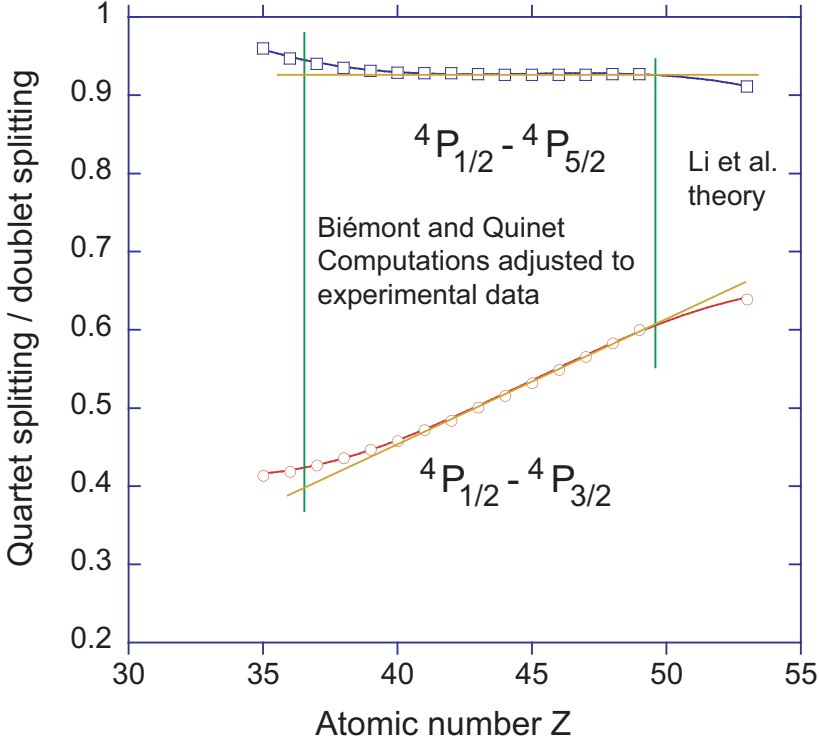

**Figure 6.** Relative fine structure splitting of the $4s4p^2\ ^4P$ term and the $4s^2 4p\ ^2P^o$ ground term in Ga-like ions. For the ground term fine structure, the experimental data compilation by Curtis and Reader [92] has been fitted by a polynomial and extrapolated. For the quartet term, the semi-empirically adjusted computation by Biémont and Quinet [69] is depicted ($Z = 37$ up to $Z = 49$); the entries for $Z = 53$ represent the computation by Li et al. [57]. The straight curry-color lines approximate the experimental data in the center part of the figure. The isoelectronic data including those for low-$Z$ and for $Z = 53$ have been approximated by third-degree polynomials (blue and red traces, respectively). The curvature at the high-$Z$ end of both trends is not evident in the data below $Z = 50$, but apparently entirely caused by a small inconsistency between the experimental data (for $Z \leq 49$) and computation (for $Z = 53$).

For Figure 6, the computed isoelectronic data on the fine structure intervals of the upper term, $4s4p^2$ $^4P$, for all measured elements up to iodine have been approximated by third-degree polynomials. The curvature of the fit functions at high $Z$ is largely determined by the single entries for $Z = 53$ and may be caused by computational differences, as well as by true atomic structure effects, and thus carries considerable ambiguity. A further extrapolation to Xe ($Z = 54$), by just one unit of charge, would be fraught with an uncertainty on the order of 1 to 2%. Neither the Berlin EBIT work on Xe [60], nor a recent NIST study [99] cover the wavelength (and in the latter case, charge state) range of present interest. Although the Xe spectrum from the Livermore EBIT shows spectral line features close to the wavelengths indicated by such an extrapolation, several candidates are available in each case within the statistical uncertainty range. Hence, the extrapolation of the experimental data in this case falls short of the accuracy required for a meaningful result. Next, we look for guidance from atomic structure computations.

Hu et al. [96] have computed low-lying levels in relatively-low charge Ga-like ions, from Ge II to Nb XI. In the lighter ions, their $4s4p^2$ $^4P_J$ levels differ from experiment by several percent, whereas for the $4s^24p$ $^2P^o_{3/2}$-$4s4p^2$ $^4P_{5/2}$ transition wavelength in Zr X and Nb XI, the deviation is only 0.2%. The computations laudably extend all the way to U ($Z = 92$), but for the high-$Z$ range, only sample results of a few levels are offered in the publication. Singh et al. [100] have performed MCDF computations on Ga-like Ba ($Z = 56$), with an emphasis on transition rates. They compare their atomic structure results to an experiment by Reader et al. [98], which includes wavelengths of two of the five $4s^24p$ $^2P^o_J$-$4s4p^2$ $^4P_{J'}$ transitions. These permit one to extend the data range for one of the two sets in Figure 6 and reveal that neither the theoretical results [57], nor the measurements [56] for iodine are fully compatible with the extrapolated trend of the lower-$Z$ experimental data (which are shown in Figure 6). According to the trends shown in Figure 6, the 3/2-3/2 transition in the Ga-like iodine ion ought to have a longer wavelength than reported previously [56] and should therefore be assigned to a weaker spectral structure than assumed so far (for example, at $30.75 \pm 0.02$ nm). The upper level of this transition is longer-lived than the other $4s4p^2$ $^4P_J$ levels by about one order of magnitude, which in the time-resolved beam-foil spectroscopic observations translates into a signal that is weaker also by an order of magnitude, compared to the other two levels of the same term. Further measurements of just one component of the intercombination multiplet, the $4s^24p$ $^2P^o_{3/2}$-$4s4p^2$ $^4P_{5/2}$ transition wavelength, have been reported for Sm ($Z = 62$) and Er ($Z = 68$) [101], and so on. Those rare earth elements lie beyond the present range of interest. Biémont et al. [102] have published an extension of their earlier study of Ga-like ions, which had reached up to In ($Z = 49$); however, the new MCDF work begins only at $Z \leq 70$, and thus, it leaves a wide gap to bridge.

In a different approach, Figures 6–8 display the deviations of various computational results from those obtained by Santana using both an (*ab initio*) MRMP approach [13] (some very recent MBPT computations by the same author, with and without configuration interaction (CI), corroborate the results of those computations and come even closer to the experimental data, but have not yet been expanded to as many elements). In these two figures, the semi-empirically-adjusted Hartree-Fock with statistical exchange and relativistic corrections (HXR) work by Biémont [69] also represents the experimental data at lower $Z$. Unfortunately, the experimental data for $Z = 53$ [56] are not accurate enough to mark a decisive trend. The various (*ab initio*) MCDF computations [57,96] scatter by several percent from each other. The curvatures of the isoelectronic trends of such computational results indicate that such computations converge better at higher values of $Z$. We also note that the trends of the results obtained by Hu and Yang [96] (relative to our reference computation and the experimental data) are not smooth, which suggests shortcomings in the computations and a limited reliability of the results. Clearly, experiment is needed to find out which *ab initio* computation is best and if any of the present sample may be considered accurate enough for practical purposes. In terms of predictive power, only long series of isoelectronic atomic structure computations (of the present sample, such as those by Santana [13]) are worth checking against the reliable measurements at low $Z$ (where the semi-empirically-adjusted computation by Biémont and Quinet [69] succeeds).

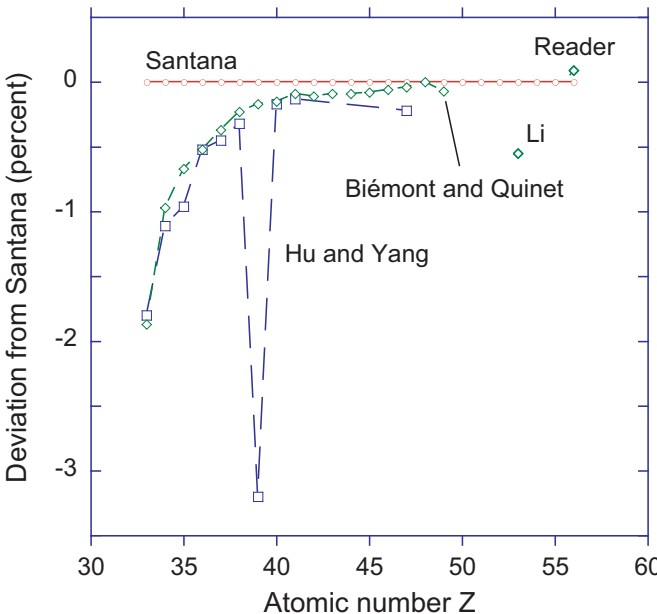

**Figure 7.** Deviation of the $4s^2 4p$ $^2P^o$ ground state fine structure in Ga-like ions from the MRMP computational results obtained by Santana [13] (open circles at zero line). The Biémont and Quinet [69] data have been semi-empirically adjusted to the low-*Z* experimental data. Evidently, the multi-configuration Dirac–Fock (MCDF) calculations by Hu and Yang [96] are close to the isoelectronic trend of the experimental data, but for $Z = 39$, they deviate markedly, which jeopardizes their reliability. Li [57] MCDF result; Reader [98] experimental data point.

## 3.4. Ge-Like Ions

In Ge-like ions, the ground configuration is $4s^2 4p^2$, and the first two excited configurations are $4s4p^3$ and $4s^2 4p4d$. The ground configuration has five levels $^3P_{0,1,2}$, $^1D_2$ and $^1S_0$. The lowest level of a parity opposite to that of the ground state is $4s4p^3$ $^5S^o_2$. This level corresponds to the $2s2p^3$ $^5S^o_2$ level in C-like ions and the $3s3p^3$ $^5S^o_2$ level in Si-like ions. The intercombination decays of these quintet levels to the $ns^2 np^2$ $^3P_{1,2}$ levels of the ground term have been seen long ago in astrophysical spectra of the first few ions of the C I and Si I isoelectronic sequences. A decay to the $ns^2 np^2$ $^1D_2$ level is possible, too, but is expected to be rather weak. Progress beyond those first few ions has been held up for many years by computational accuracy problems that rendered predictions unreliable; the $^5S^o_2$ level mixes with the $^3P^o_2$ level, as well as with the $^1D^o_2$ level of the same configuration. Those levels in turn feature decay channels with cancellation effects that influence the $^5S^o_2$ level decay rate. For more detail on such calculations, see [103,104]; for results of beam-foil measurements on Si-like ions (and for more references), see [8,105]. Considering the enormous problems with determining the $3s3p^3$ $^5S^o_2$ level and its decay rates in Si-like ions, one wonders about the situation in Ge-like ions and the meaning and merit of some literature data. The prediction of the branch fractions (similarly the line ratio) of the two transitions has been a challenge in Si-like ions (where it varies along the isoelectronic sequence, depending on the multi-level mixing); this is likely not simpler to achieve with accuracy in Ge-like ions with their more than twice as many electrons.

The decays of the $4s4p^3$ $^5S^o_2$ level to the $4s^2 4p^2$ $^3P_{1,2}$ level in Ge-like ions are expected to show as two spectral lines longward (but not far from) the strongest of the intercombination transitions in Ga-like ions, $4s^2 4p$ $^2P^o_{3/2}$-$4s4p^2$ $^4P_{5/2}$, which in turn appear longward (but not far from) the intercombination transition in the spectra of Zn-like ions. However, this rule of thumb is clearly not precise enough to yield identifications, not even when combined with a line spacing that corresponds to the $4s^2 4p^2$ $^3P_{1,2}$ level spacing of the ground term. The fine structure splittings of the ground term have been compiled (for elements from $Z = 36$ up to $Z = 48$) by Litzén and Reader [67] and by Litzén and Zeng [106]. For five elements ($43 \leq Z \leq 47$), Biémont et al. [70] have performed relativistic

Hartree–Fock computations with adjustable Slater parameters; the results of this adjusted calculation corroborate the internal consistency of the experimental analysis by the other authors. However, Figure 9 reveals that the semi-empirical adjustments worked out differently among the two author groups, and their predictions for the 4s4p$^3$ $^5$S$^o_2$ level do not link up.

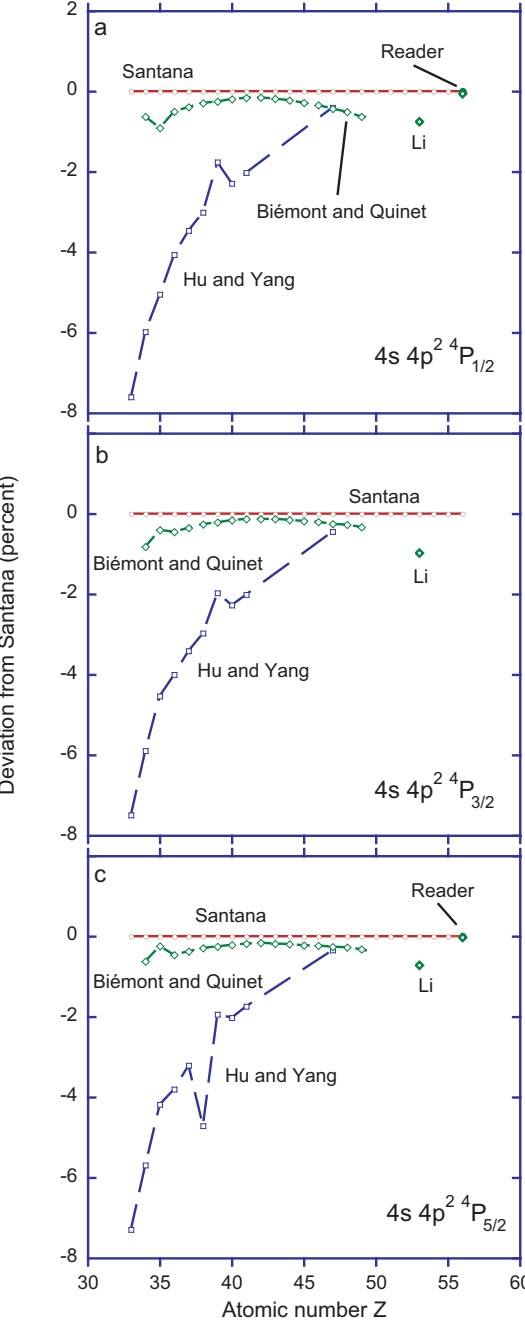

**Figure 8.** Deviation of the 4s4p$^2$ $^4$P$_J$ levels in Ga-like ions (**a**) *J*=1/2, (**b**) *J*=3/2 and (**c**) *J*=5/2 from the MRMP computational results obtained by Santana [13] (open circles at the zero line). These computational results are close to the results obtained by Biémont and Quinet [69], who adjusted their HXR computational results semi-empirically to the trend of the experimental data. The MCDF results by Hu and Yang [96] clearly deviate from experiment at low *Z* and approximate the latter only for higher values of *Z*. Li [57] denotes MCDF computations, as well, while Reader [98] represents experimental data.

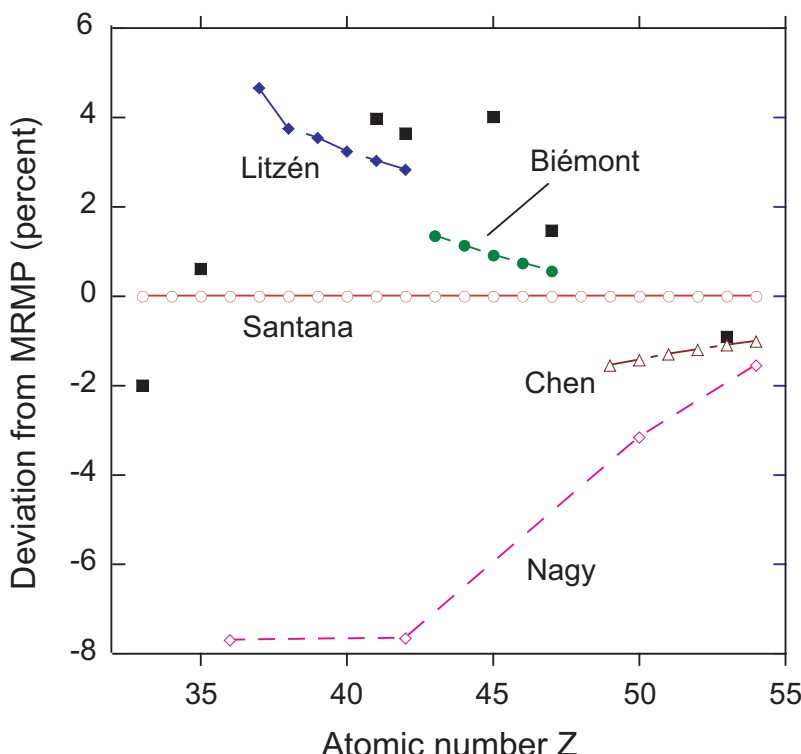

**Figure 9.** Deviation of the $4s4p^3$ $^5S^o_2$ level in Ge-like ions from the MRMP computational results obtained by Santana [13] (open circles at the zero line). Black squares refer to experiments [21,48,51–53,55,56,107] (for As and Br, the error bars are smaller than the symbol size; for $Z \geq 41$, all experimental entries are from more or less tentative line identifications and thus depicted without error bars); all other entries refer to computations: Litzén and Reader (blue diamonds) [67] and Biémont et al. [70] (full green dots) are based on semi-empirically-adjusted Cowan code calculations (apparently with different adjustments, which cause the discontinuity of the isoelectronic trends between the two datasets); Nagy and El Sayed (open red diamonds) [68] and Chen and Wang (open mauve triangles) [72] use *ab initio* MCDF computations.

In the NIST ASD database, a measured $4s4p^3$ $^5S^o_2$ level value is given for As II and Br IV, but not for Se III or Kr V. The corresponding database entries (given in parentheses) for Rb VI to Mo XI refer to work by Litzén and Reader [67], who list numbers for this level position that they have derived indirectly from their atomic structure computations and spectrum analysis. Their analysis is supported by Cowan code computations with adjustable Slater parameters, adjusted so that recognized spectral features are best reproduced. In this case, they thus established estimates of the quintet level position without having actually identified any spectral lines that link to this level in their data. Considering the problems encountered by Ellis and Martinson [103,104] and many others when trying to compute the quintet level position in ions of the C I and Si I isoelectronic sequences accurately, the estimate by Litzén and Reader may be a useful pointer, but it is of very limited predictive and descriptive power. If one tries to see a common isoelectronic trend of the measured data for the quintet level in As II and Br IV and the estimated values for Rb VI through Mo XI, there appears to be a discontinuity of several percent between Br and Kr. For Kr V with the $4s^2 4p^2$ $^3P_{1,2}$-$4s4p^3$ $^5S^o_2$ lines expected near a wavelength of 100 nm, this corresponds to a wavelength uncertainty of several nanometers, that is an interval in which likely several lines can be found that lack further identification features. Litzén and Zeng [106] have extended the experimental and analytical work of Litzén and Reader towards higher $Z$, but without mentioning the quintet level of present interest.

The most complete EUV spectrum of a Ge-like spectrum of a multi-charged ion appears to be Br IV [107,108], an analysis of which was published more than four decades ago. Beyond that, there are

spectra that cover incidental sections of the EUV spectral range and some transition arrays [60,109–116], but the coverage is far from complete. This incompleteness is to be expected for the very many open-shell levels, but it also applies to something as close to fundamental as the fifth level above the ground level, the lowest one of opposite parity, which apparently is not even mentioned in those tabulations.

Concerning atomic structure computations, the literature is comprised of a number of systematic studies that in combination provide a reasonable coverage of wide parts of the Ge isoelectronic sequence. Biémont et al. [70] apply (semi-empirically adjusted) relativistic Hartree–Fock computations to elements from $Z = 37$ (Rb) to $Z = 47$ (Ag), but the quintet level of present interest is listed only for ions from Tc ($Z = 43$) (an element for which no experimental data are expected in the foreseeable future) to Ag ($Z = 47$). Nagy and El Sayed [68] treat Ge-like elements with $Z = 36, 42,$ 50 and 54 by MCDF computations and include the $4s4p^3\ ^5S_2^o$ level. Their $4s^24p^2\ ^3P_1$ level position in Kr differs by 10% from the experimental value listed by Saloman [17]. Their result for the $4s4p^3\ ^5S_2^o$ level is some 13% lower than the experiment-guided estimate by Litzén and Reader [67]. At higher $Z$, the MCDF results obtained by Nagy and El Sayed appear to become more meaningful; for Xe ($Z = 54$), they approach the MCDF and MBPT results obtained by Chen and Wang [72] within a couple of percent. Chen and Wang [72] cover the elements from $Z = 49$ (In) to $Z = 58$ (Ce) by MCDF and (for $Z = 54$ only) MBPT computations. The two schemes yield results that for Xe come within a small fraction of one percent of each other (and probably also with experiment; see below). For I ($Z = 53$), there also is close agreement with the MCDF results obtained by Li et al. [57]; their results also come close to experimental data on Zn- and Ga-like ions. Palmeri et al. [117] apply MCDF computations to highly-charged Ge-like ions of elements $Z = 70$ and higher, that is in a range beyond our present range of interest.

The numbers resulting from the various computations are difficult to visualize in an isoelectronic plot, because they are of different origin and span a considerable range. For example, the $4s4p^3$ $^5S_2^o$ level values associated with most of the experiments and some of the semi-empirically-adjusted computations involve fine structure levels of the ground configuration that have been obtained in other experiments. Hence, we split the visualization of the information in several ways. One way is the actual level value delivered by experiment or computation, but only relative to a reference computation; the other is the predicted pair of intercombination decays in the experimental spectra. In Figure 9, we present the deviation of the claimed $4s4p^3\ ^5S_2^o$ level values just discussed from the results of MRMP computational results obtained by one of us [13]. Thus, most of the atomic structure is already taken care of, and the graphical presentation deals with the differences among the computed results and the actual data. Of course, calculations for several elements yield results that provide an isoelectronic trend for the elements covered; however, in the present case, the individual ranges covered in the literature are deplorably short, and the various computations do not link up to a common trend. The scatter of the results reveals that the calculations so far offer a rather limited predictive value. There are seven experimental data points, all with their specific problems. The low-$Z$ experiments using classical spectroscopy offer the highest accuracy, but near the neutral end of the isoelectronic sequence, the Hylleraas expansion of atomic parameters may be different from the high-$Z$ trends. For low-charge ions, the computations face particular problems with the relatively weak central field; hence, a comparison of theory and experiment is shaky in this range. The five beam-foil data points have the advantage of isotopic purity, but the spectral resolution is poorer and the signal rate low. There is no published evidence from the spectra of other light sources yet.

We present the published predictions of the two intercombination decays inserted into the beam-foil spectra of Figures 1–3. In the spectra of Nb (Figure 1, top and middle panels), the positions of the two $4s^24p^2\ ^3P_{1,2}$-$4s4p^3\ ^5S_2^o$ lines predicted by Litzén and Reader [67] do not match any notable spectral features. However, at slightly shorter wavelengths, at least a pair of very weak lines fits the expected wavenumber difference [48]. Similarly, in the spectrum of Mo (Figure 1, bottom panel), the prediction based on Litzén and Reader [67] is not corroborated, but at somewhat

shorter wavelengths (52.23 nm and 54.96 nm, respectively), there are moderately weak lines that fit the appropriate wavenumber difference. In both cases, the quintet level would lie higher than estimated by Litzén and Reader by about 1700 cm$^{-1}$. Unfortunately, this is the end of the elements for which Litzén and Reader predict the quintet level of interest. For the next five elements, Biémont et al. [70] provide numbers (also from semi-empirically-adjusted computations) that we can compare with the spectra of Rh and Ag (Figure 2). In the spectrum of Rh (Figure 2, upper panel), the prediction matches two moderately weak lines, one of which is blended with one of the components of the intercombination transition multiplet in the Ga-like ion. However, there is another match at somewhat shorter wavelengths (41.51 nm and 43.74 nm, respectively), which would imply that the quintet level of interest lies about 5000 cm$^{-1}$ higher than predicted by Biémont et al. This same offset would fit the spectra of Ag (Figure 2, middle and bottom panel), with lines at 36.55 nm (blended) and 38.66 nm, respectively. In this case, again, there are no suitable lines at the position predicted by Biémont et al. [70] (the difference in offset corresponds to the mismatch of the isoelectronic trends of the studies by Litzén et al. and by Biémont et al., which is evident in Figure 9) The spectra in Figure 3 are of iodine ($Z = 53$) (upper panel, beam-foil spectroscopy [56]) and Xe ($Z = 54$) (lower panel, data from an electron beam ion trap [71]), respectively. The only comparison available for the Ge-like spectra of these two elements is with theoretical work, with two studies addressing a single element each in the neighborhood, and only the study by Chen and Wang [72] indicating isoelectronic trends (but falling short of any overlap with the lower-$Z$ systematic work mentioned above). Evidently, the convergence of the atomic structure program packages is good for such high charge states, and the results obtained by the various MCDF and MBPT computations for I and Xe are close to each other. In iodine, these results suggest that again one of the two decay branches of the quintet level is blended with the strongest (and much stronger) line of the intercombination transition multiplet in the Ga-like ion. In the electron beam ion trap spectrum of Xe, the computations all point to a short wavelength range (25.7 nm to 27.2 nm) with many lines. Unfortunately, line blends are likely, and without recognizable distinguishing features, possibly available at higher spectral resolving power, no individual lines in the measured spectrum can be properly identified with the two quintet level decays of present interest. In contrast, in the beam-foil work, decay curves have been recorded that corroborate the candidate status of the marked lines with (often blended) intercombination transitions; moreover, where both lines are seen, the line ratio slightly favors the longer-wavelength decay to the $J = 2$ level over that to the $J = 1$ level.

In (partial) conclusion, the Ge-like ion levels and lines of present interest have not yet been sorted out conclusively, since the two lines of somewhat uncertain predicted wavelengths, spacing and relative intensity are not easily recognized unambiguously. None of the *ab initio* computations so far can prove its merit (in our $Z$ range of interest) by a successful match with experiment or by intrinsically proving spectroscopic accuracy, especially in the lower-$Z$ range. Thus, the apparently dominant problem here seems to be not relativity or QED, but computational convergence at low core charge values. What is needed is a consistent calculation of the ground state fine structure, as well as the $4s4p^3$ $^5S^o_2$ level along a long section of the isoelectronic sequence, beginning at low Z, as has been demonstrated preliminarily by Santana [13].

## 4. Discussion and Outlook

Both better measurements, as well as data on more elements are required to improve the collection of reference data. A possible way forward towards a more complete term analysis of many elements and ions might employ the systematic electron beam energy variation in an electron beam ion trap, as done in the Biedermann study of Xe [60]. In this way, sufficiently small energy increments (of, say a few dozen eV) result in spectra in which a single charge state is added to a spectrum of many lower charge state ions, and thus, from a few dozen spectra, one can derive which ions cause which lines, which is a major handle in the process of line identification. Some such work at NIST has concentrated on elements just above the range of elements discussed here (for example, see [98,101]).

A typical wavelength range covered there was 4 to 20 nm, which partly leaves out the intercombination transitions discussed here (for higher-*Z* elements, the intercombination lines of present interest would all fall into the wavelength range in this spectrometer system setting). Interestingly, an observation of what in beam-foil spectra regularly appears as the strongest transition of the intercombination transition multiplet in Ga-like ions has been claimed for several elements in the NIST electron beam ion trap, although this line cannot be as outstandingly bright in the low-density environment as it appears after foil excitation, but none of the other four lines of the same multiplet have been identified there.

Another way, exploiting the lower decay rates of intercombination transitions, might use beam-foil spectroscopy and the technique of recording delayed spectra (demonstrated and discussed here), in which the high number of contributions of short-lived levels and their decays is reduced, so that the remaining spectra are more recognizable in comparison to suitable computations. However, nowadays, there are few accelerator laboratories left that can provide sufficient ion beam currents of the elements of interest at the appropriate ion energies to produce the ions in the proper charge states.

All of this would be greatly helped by accurate computations, if available, and the accuracy of those would best be corroborated by comparison with experimental data. Better calculations are wanted for ions with more than a single electron in the valence shell. An obvious problem with several of the existing calculations is that they do not cover all elements; serial publications on short segments of isoelectronic sequences are counter-productive, because for the isoelectronic trends to be usefully employed, there ought to be the option to compare prediction with measurement, preferably for several elements. While atomic structure codes provide output for a great number of levels of a given ion, there is little merit in putting ever so many levels into print, because it is not realistic to assume that most of the high-lying levels will ever be explicitly needed for comparison with experimental data. For spectrum simulations based on large numbers of levels, everybody would try to do their own computations and electronically transfer the bulk of the output to the next processing stage rather than reading a journal and retyping all the numbers. It would be more sensible for the providers of computed atomic data to demonstrate the quality of their work by concentrating on low-lying levels and on those few that have a chance to be measured eventually. Proving the accuracy of one's techniques (that is, demonstrating quality assessment) is much more meaningful than printing many numbers of unknown reliability. Last, but not least, for computations that are published only after experimental data have become available, the chance to display predictive power has been missed.

**Author Contributions:** Conceptualization, E.T., P.Q. and P.P. Data curation, E.T.; Investigation, E.T., J.A.S., P.Q. and P.P. Methodology, E.T. Project administration, E.T. Software, J.A.S. Visualization, E.T. Writing the original draft, E.T. Writing review and editing, E.T., J.A.S., P.Q. and P.P.

**Acknowledgments:** Part of this work was performed under the auspices of the U.S. Department of Energy by the Lawrence Livermore National Laboratory under Contract DE-AC52-07NA27344.

**Conflicts of Interest:** The authors declare no conflicts of interest.

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
