# Peer review of "Intercombination Transitions in the n = 4 Shell of Zn-, Ga-, and Ge-Like Ions of Elements Kr through Xe"

_atoms, doi:10.3390/atoms6030040_

Round 1

Reviewer 1 Report

Reviewer comments

Manuscript ID: atoms-303759

Title: Intercombination transitions in the n=4 shell of Zn-, Ga-, and Ge-like ions of elements Kr through Xe

Authors: Elmar Träbert *, Juan A. Santana, Pascal Quinet, Patrick Palmeri

Sp.Issue: Current Developments and Applications of Atomic Structure and Radiative

Process Investigations

The manuscript is clearly written and the material is interesting for potential readers. But i
t is necessary to make some small corrections and improvement of the paper. I would like to recommend publications of this paper in Atoms but with few requests.

Some requests:

I suggest to the authors to change the type of the manuscript from Article to Review.  As they say in the Abstract (line 5) “We review the (mostly computational) progress since made and find that a consistent set of state-of-the-art computations of Ga- and Ge-like ions, for the elements from ….”  and on last paragraph in Introduction “The present study therefore does not present a

new idea, but applies the essence of the earlier findings ...”

In this context it would be good if the authors describe a little bit more about the current state in the experimental studies. Briefly.

-Figs. 1 and 2, should be inserted into the text close to their first citation and must be numbered following their number of appearance. Currently, Figs 1 and 2 are mentioned on page 13 after Figs 3,4, 5 ….

- page 9, Fig. 3 upper panel: Träbert 2010, should be =>  Träbert 2011 (Eq. 56)

- page 11, Fig. 4, Kim et al.=> Kim et al. 1991.

- page 2, caption of the Fig 9.   “… Black squares refer to experiments cited in the text …” =>        “… Black squares refer to experiments cited in the text [Refs.] …”

For a 27-page article the Discussion is very short. It will be good for potential readers if the authors extend description. ..about problems in experiment and theory or similar.

In conclusion, the article should be accepted.

Respectfully,

Author Response

The manuscript is clearly written and the material is interesting for potential readers.
But it is necessary to make some small corrections and improvement of the paper.
I would like to recommend publications of this paper in Atoms but with few requests.

Some requests:

I suggest to the authors to change the type of the manuscript from Article to Review.  
As they say in the Abstract (line 5) “We review the (mostly computational) progress since
made and find that a consistent set of state-of-the-art computations of Ga- and Ge-like ions,
for the elements from ….”  and on last paragraph in Introduction
“The present study therefore does not present a
new idea, but applies the essence of the earlier findings ...”

Reply: Thanks to the reviewer for a careful and in the end positive assessment of our manuscript.
The clerical mistakes have been repaired. The reviewer suggests to change the manuscript classification from Article to Review. We have pondered both options before submission, as there are various arguments either way. We preferred the one option, whereas the reviewer prefers the other.
The extensive Santana computations used for the figures on systematics are unpublished at source and point way beyond a review. Moreover, it is uncommon for reviews (though not unheard of) to make detailed intercomparisons of results and reinterpretations of published data.

While we share the reviewer's point that our manuscript is somewhere between a conventional research paper and a review, we authors see it closer to the former than to the latter.

However, we will confer with the Special Issue Editor on this matter and then find a solution.

In this context it would be good if the authors describe a little bit more about the current
state in the experimental studies. Briefly.

Reply: Thanks for the hint. The current state of experimental work is abominably patchy.
Practically all worth saying has been said in our manuscript (which we have pondered again, all the text). A sad point not expanded upon is the "biological" demise of beam-foil spectroscopy by the retirement of facilities and practitioners. However, a surprising detail from this investigation is the recognition that the databases are so poor even for low charge state ions.  Their spectra must be accessible even to moderate power laser plasma sources - if people would care and be funded ...

-Figs. 1 and 2, should be inserted into the text close to their first citation and must
be numbered following their number of appearance. Currently, Figs 1 and 2 are
mentioned on page 13 after Figs 3,4, 5 ….

Reply: The reviewer has overlooked that figures 1 to 3 are first mentioned on page 6 and begin on page 7.

 - page 9, Fig. 3 upper panel: Träbert 2010, should be =>  Träbert 2011 (Eq. 56)

 - page 11, Fig. 4, Kim et al.=> Kim et al. 1991.

 Reply: Another reviewer and the staff editor have reminded us that years with the author names are  "not wanted" in this journal, and we are unifying our text and figures to avoid this break of style prescription.

 - page 2, caption of the Fig 9.   “… Black squares refer to experiments cited in the text …” =>        
 “… Black squares refer to experiments cited in the text [Refs.] …”

 Reply: O.k., changed

 For a 27-page article the Discussion is very short. It will be good for potential readers if the
 authors extend description. ..about problems in experiment and theory or similar.

 Reply: The paper covers three isoelectronic sequences with different features and problems.
 Therefore most of the discussion is done within the three individual sections.

In conclusion, the article should be accepted.

Reply: Thanks!

Reviewer 2 Report

The authors present a compilation of experimental and theoretical data on a particular set of atomic transitions in Zn-, Ga-, and Ge-like ions. It is found that the various calculations are significantly inaccurate in reproducing the experimental line positions which were mainly obtained from beam-foil spectroscopy partly already some decades ago. It is concluded that more diligent and systematic calcuations are required to improve the situation.The notion, that atomic structure calcuations perform poorly on many-electron systenms is by no means new. The motivation for studing in particular  Zn-, Ga-, and Ge-like ions remains unclear despite the lengthly introduction. In my opinion the paper contains too little new physics to warrant publication as a regular research article. It would probably be acceptable as a conference paper.

Author Response

The authors present a compilation of experimental and theoretical data on a particular
set of atomic transitions in Zn-, Ga-, and Ge-like ions. It is found that the various calculations
are significantly inaccurate in reproducing the experimental line positions which were mainly
obtained from beam-foil spectroscopy partly already some decades ago. It is concluded that
more diligent and systematic calcuations are required to improve the situation.The notion,
that atomic structure calcuations perform poorly on many-electron systenms is by no means new.
The motivation for studing in particular  Zn-, Ga-, and Ge-like ions remains unclear despite the
lengthly introduction. In my opinion the paper contains too little new physics to warrant publication
as a regular research article. It would probably be acceptable as a conference paper.

Reply:
We are sorry that the reviewer judges our manuscript so poorly. Our paper does not aim at "new physics", but at checking the validity of existing physics treatments, both in experiment (spectroscopy) and computation. We consider the quest for "new physics" idle, if the quality of a significant fraction of the "existing physics" is so uncertain.
Our manuscript also shows that not all new computational work is up to the task. Surely all
authors see their own computational work as diligently conceived and executed, although most of them seem unable to come up with a valid quality assessment. We therefore beg to differ from the
views of the expert reviewer in that we try to find out what work appears to be reliable and how far that carries. We also trust that all this explanation is, in fact, in our text and can be recognized by interested readers.
Indeed, the recognition that computation has problems with many-electron ions
is not new (we authors have demonstrated such cases before), but the present examples have not been
scrutinized before, yet they systematically extend our earlier studies.

Reviewer 3 Report

The English is, in places,  unnecessarily cumbersome

We have selected a group of available wavelength measurements (in the extreme-ultraviolet wavelength (EUV) range) on ions with one, two, three, or four electrons outside a closed-shell electron core, in the middle of the range of natural elements, say, from Kr (Z = 36) to Xe (Z = 54) (an arbitrary choice in the present context), in order to test the accuracy of certain atomic structure calculations versus a systematic progression of complexity.

There are many more examples like this.

I have very mixed feelings about this manuscript. I agree very much with the criticism the authors have about calculational papers presenting results that can never be tested instead of concentrating on low lying levels when experimental tests are possible. It is not enough to show that if two theoretical methods give similar results then that is evidence for correctness, they could both be missing (or underestimating/overestimating) a similar interaction. This has been seen before. This does need to be (re-) pointed out.  

For example: In computation, resources have expanded massively, and one wonders why so many recent computations address only short sections of isoelectronic sequences (maximising the number of publications instead of their usefulness?). Since the results of one computation rarely match the results of another computation for the same atomic system, it is desirable to compare the results to those of measurements - but with computations for individual ions or only short segments of isoelectronic sequences, there rarely is overlap among computations.

It also very good that the authors use iso-electronic sequences and so-called obs-calc (or calc-calc) plots to emphasis deviations or agreements in available data, this concept seems to be losing ground. 

On the other hand, maybe it is not really feasible for any spectroscopy group to re-visit the old testing grounds if intercombination, IC, lines, as suggested many times in this manuscript. Although it is correct that there are oceans of missing data, both for the iso-electronic squences discussed here, and many others, there are so many fewer spectroscopy groups operating now. It is does not really seem conceivable that any surviving group will go back to re-visit IC lines. My guess is that fusion diagnostics have what they need as do astrophysicists, or at least they have got basically what they are going to get. In earlier days, there were numerous light sources where IC lines could be studied, at the authors mentioned, beam-foil spectroscopy and spectroscopy at magnetic confinement fusion plasma devices. Now-days, beam-foil spectroscopy is basically extinct and fusion laboratories have heavy programs and usually no resident spectroscopists. There are very few Electron Beam Ion Trap facilities left dealing purely with spectroscopy, mainly due to funding problems I imagine, and I cannot see this situation getting much better. Also, is it ethically correct for authors to tell (suggest) what other researches should do so that they can satisfy their curiosity, this is only my thought. Sadly, as the problems associated with IC lines are interesting, it is difficult to see who would work to “put these problems right”. 

 Experimental spectroscopy studies are getting very few and far between and usually have to motivate the cost through either fusion, astrophysics or light source for future lithography, so studies of pure atomic structure, although badly needed, are not common, as I am sure the authors are fully aware of. However, it is important to point out that the garden of known atomic data is not a rosy one. Many are under the impression that everything, or at least a lot, is known whereas the truth is far from this.

I can accept the manuscript for publication if the authors clear up some of the cumbersome sentence structures.

Author Response

The English is, in places,  unnecessarily cumbersome

We have selected a group of available wavelength measurements (in the extreme-ultraviolet
wavelength (EUV) range) on ions with one, two, three, or four electrons outside a closed-shell
electron core, in the middle of the range of natural elements, say, from Kr (Z = 36) to Xe
(Z = 54) (an arbitrary choice in the present context), in order to test the accuracy of certain
atomic structure calculations versus a systematic progression of complexity.

There are many more examples like this.  

Reply: Thanks for the reviewer scrutiny, thoughtful comments, and suggestions.
My (ET) experience from decades of refereeing is the frequent appearance of heavy English
loaded with impressive physics - and being so garbled, misguided, or unintelligible that
a reviewer is taken aback and has to struggle to find out whether there is any consistency
in a given manuscript. I am happy to see this reviewer's much milder criticism of our manuscript,
and we try again to decomplexify (what a word!) our text.

I have very mixed feelings about this manuscript. I agree very much with the criticism the
authors have about calculational papers presenting results that can never be tested instead
of concentrating on low lying levels when experimental tests are possible. It is not enough to
show that if two theoretical methods give similar results then that is evidence for correctness,
they could both be missing (or underestimating/overestimating) a similar interaction.
This has been seen before. This does need to be (re-) pointed out.  

For example: In computation, resources have expanded massively, and one wonders
why so many recent computations address only short sections of isoelectronic sequences
(maximising the number of publications instead of their usefulness?). Since the results of one
computation rarely match the results of another computation for the same atomic system,
it is desirable to compare the results to those of measurements - but with computations for
individual ions or only short segments of isoelectronic sequences, there rarely is overlap among computations.

Reply: We have reworded this paragraph of the introduction to enhance this important point, the
need to establish measures of reliability (accuracy) in atomic structure computations. Phys Rev A
these days requires such explanations by authors who submit theory articles. This is a very
helpful move which unfortunately has not been taken up by many other journals yet. Thus we still
see publications with very many numbers, but without substantiation.

It also very good that the authors use iso-electronic sequences and so-called obs-calc (or calc-calc)
plots to emphasis deviations or agreements in available data, this concept seems to be losing ground.

 On the other hand, maybe it is not really feasible for any spectroscopy group to re-visit the old
testing grounds if intercombination, IC, lines, as suggested many times in this manuscript.
Although it is correct that there are oceans of missing data, both for the iso-electronic squences
discussed here, and many others, there are so many fewer spectroscopy groups operating now.
It is does not really seem conceivable that any surviving group will go back to re-visit IC lines.
My guess is that fusion diagnostics have what they need as do astrophysicists, or at least they
have got basically what they are going to get. In earlier days, there were numerous light sources
where IC lines could be studied, at the authors mentioned, beam-foil spectroscopy and spectroscopy at magnetic confinement fusion plasma devices. Now-days, beam-foil spectroscopy is
basically extinct and fusion laboratories have heavy programs and usually no resident
spectroscopists. There are very few Electron Beam Ion Trap facilities left dealing purely
with spectroscopy, mainly due to funding problems I imagine, and I cannot see this situation
getting much better. Also, is it ethically correct for authors to tell (suggest) what other researches
should do so that they can satisfy their curiosity, this is only my thought. Sadly, as the problems
associated with IC lines are interesting, it is difficult to see who would work to “put these problems right”.

 Experimental spectroscopy studies are getting very few and far between and usually
 have to motivate the cost through either fusion, astrophysics or light source for future lithography,
 so studies of pure atomic structure, although badly needed, are not common, as I am sure the
 authors are fully aware of. However, it is important to point out that the garden of known atomic
 data is not a rosy one. Many are under the impression that everything, or at least a lot, is known
 whereas the truth is far from this.

 I can accept the manuscript for publication if the authors clear up some of the cumbersome sentence structures.

Reply: Many, if not all, of the reviewer's thoughts mirror our own. Thanks for letting us know that we are not alone.
The reviewer ponders whether it is worth lamenting about the sad state of the garden of known atomic data, if hardly anybody listens or adepts are around to take up the tedious task of laboring towards remedy. Of course, much of this boils down to funding, and funding is hard to come by without extensive lobbying. The pursuit of reliable atomic data is not self-explicable to all envisaged supporters. The misconception of "nothing new in atomic data physics, all is well under control and computable to sufficient accuracy" is widespread even among physicists.
For low charge states (the most likely in our environment), the ultimate goal of much of atomic structure computation, the task is daring, and isoelectronic comparisons largely fail (as Edlén in his handbook article has demonstrated half a century ago). For highly charged systems, the central Coulomb force dominates, and computations converge much more rapidly.
However, here theory struggles with relativity and QED, both of which carry prominence in prestigeous journals. I recall an early paper by (highly esteemed, for good reason) Walter Johnson which (later) badly matched experiment, because QED was missing (and not even mentioned). Work in that high-Z and high-charge range has improved a lot since.
Therefore we have looked at the rather non-descript range of ion charges in the middle - our manuscript shows how poor the situation is there, where QED is considered unimportant and relativity a minor contribution. Yet complexity rules, and routine application of run-of-the-mill atomic structure packages remains insufficient. Unfortunately this situation needs to be retold, since
producers of computer-generated numbers have a tendency to not quantify their persistent problems and lack of accuracy.
Meanwhile the NIST ASD online database has recently (partly) been flooded by computed entries while there is no mechanism to correct earlier table entries that have been recognized as very likely faulty.

Round 2

Reviewer 2 Report

In their reply the authors do not indicate that they have signifcantly improved their manuscript. They merely ask for reconsideration. Therfore, I do not see a point in reassessing the manuscript.  I rather suggest that another expert be consulted.

Academic Editor Notes
The paper is generally well written and is acceptable for publication, but only after implementing a few, minor but necessary,  changes. Some of these are of general nature related to the style of the journal and some are related to the presentation, especially the figures. Some corrections will be implemented by the editorial office during copyediting, and following is the list suggesting the changes/corrections required, but it should not be taken as exhaustive.
The authors need to go through the paper again very carefully to implement the suggested
as well as non-suggested changes of similar nature.

Author comment: Thanks to the academic editor for spotting those weak points.
We bow to the academic editor for his/her scrutiny, different perspective,
and helpful suggestions.  Perhaps not surprisingly, we do not always share the
views communicated. Some of the somewhat stilted wordings the academic editor
justly criticizes have been used, because experience tells us that some of the
prospective readers (including one of the reviewers) are less versed in the field
than the academic editor.

1) p1: It is not customary to give the Tel. No. or PACS nos.
Author comment:
As far as I can remember, the telephone number has been asked by a submission form; surely our
submitted file did not have that information of our own initiative. Sometimes the editorial office
may introduce details beyond the authors' horizon. In this case, I am happy to see the telephone
number go away again. The MDPI template is unspecific on PACS numbers.

2) p3, L3: insert moderately before heavy and in L2 of bottom para replace tungsten by 'it'
Author comment: done

3) p4, L4 of first new para: delete ions and 'on beam foil spectroscopy' from L8.
In fact, it is a common problem throughout the paper to repeatedly write the same word/s in
a single sentence which makes the reading uncomfortable and annoying.
Two other examples are in (5) and (8) below, but the list is rather long.
Author comment:  We are sorry to note the academic editor's displeasure with parts of our text.

4) p4, L14 of first new para: replace 'than low-charge state ions.' by 'those in low-charge states.'
Author comment: done

5) p4, middle para: The word computation/s appears three times in a single sentence.
I suggest the following or the authors can rewrite on similar line.

     The results of one computation rarely match the results of another for the same atomic system,
     which illustrates the limited accuracy of most atomic structure calculations.
Author comment: Done, thanks for the hint. Streamlining is welcome.  

6) p4, last line of mid para: replace 'but those experimental data often are scarce.' by 'which are often scarce.'
Author comment: done

7) p5, top para: remove bracket () around the sentence 'There are ...  [1]).' In fact on several occasions
similar writings appear which the authors should try to  minimise and preferably omit.
Author comment:  The bracket puts the enclosed text - an explanation - out of the flow.
This was done intentionally. Not changed.

8) p6, L2-4: replace 'In singly excited He-like ions, one of the electrons is a 1s electron; the active nl electron (with n
Author comment: Unfortunately, the academic editor's text was cut in transmission to us.
Our text attempts to illustrate why the problem of complexity is way beyond that of "a few electrons outside an inert closed-shell core". We have now shortened the paragraph by leaving out the example of He and by simplifying the presentation.

Author comment: Beyond the transmitted editorial comments we have also streamlined, for example, the last paragraph of the introduction. It is a pity that the academic editor's valuable notes and suggestions have not reached us in full. However, practically any text can be modified over and over again, with the hoped-for gain in clarity not necessarily staying in proportion to the effort. "Atoms" as an on-line journal relieves the pressure imposed by tight paper resources. That loosened restriction may be associated with detrimental side effects such as wordier manuscripts and lessened intellectual constraint. However, even the venerable Physical Review Letters had articles only 2 or 3 pages long when I began to work in science, and nowadays article lengths of 5 and 6 pages occur there, too.

In their reply the authors do not indicate that they have signifcantly improved their manuscript. They merely ask for reconsideration. Therfore, I do not see a point in reassessing the manuscript.
Author comment: The reviewer is mistaken. The academic editor may not have seen our extensive comments written to each reviewer report, which stated that practically all suggestions had been taken up. No, we did not disregard the reviewer effort, but adhered to it as closely as seemed ever sensible.  In my estimate we made about a hundred changes to text and figures, mostly following the reviewer guidance, suggestions, and requests.